# SAEMARK: Steering Personalized Multilingual LLM Watermarks with Sparse Autoencoders

**Zhuohao Yu**[*], **Xingru Jiang**[*], **Weizheng Gu**,
**Yidong Wang**, **Qingsong Wen**, **Shikun Zhang**, **Wei Ye**[†]
Peking University
zyu@stu.pku.edu.cn, wye@pku.edu.cn
https://zhuohaoyu.github.io/SAEMark

## Abstract

Watermarking LLM-generated text is critical for content attribution and misinformation prevention, yet existing methods compromise text quality and require white-box model access with logit manipulation or training, which exclude API-based models and multilingual scenarios. We propose SAEMARK, an **inference-time framework** for *multi-bit* watermarking that embeds personalized information through *feature-based rejection sampling*, fundamentally different from logit-based or rewriting-based approaches: we **do not modify model outputs directly** and require only **black-box access**, while naturally supporting multi-bit message embedding and generalizing across diverse languages and domains. We instantiate the framework using *Sparse Autoencoders* as deterministic feature extractors and provide *theoretical worst-case analysis* relating watermark accuracy to computational budget. Experiments across 4 datasets demonstrate strong watermarking performance on English, Chinese, and code while preserving text quality. SAEMARK establishes a new paradigm for **scalable, quality-preserving watermarks** that work seamlessly with closed-source LLMs across languages and domains.

## 1 Introduction

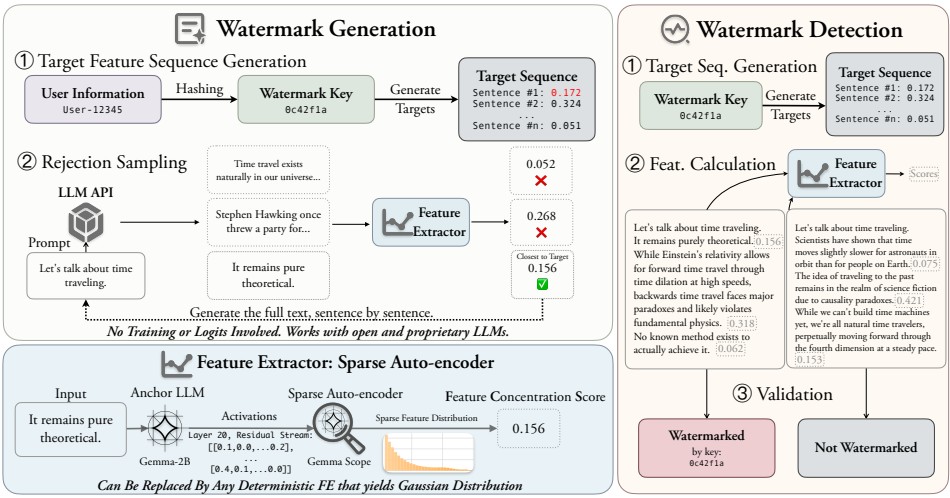

Figure 1: An overview of SAEMARK.

---

[*]: Equal contribution. [†]: Corresponding author.

39th Conference on Neural Information Processing Systems (NeurIPS 2025).

Large language models (LLMs) have revolutionized text generation across domains, from creative writing to code synthesis [1, 2]. However, their ability to produce human-quality text at scale raises serious concerns about misinformation, copyright infringement, and content laundering. As these models become ubiquitous, reliably attributing AI-generated content becomes critical for accountability and trust.

Watermarking—embedding detectable signatures into generated text—offers a promising solution. They must preserve text quality while enabling reliable detection, operate across languages and domains, and scale to distinguish between many users or sources. Most critically, they must work with real-world deployment constraints where model providers offer only API access without exposing internal parameters. The challenge becomes even more complex for *multi-bit* watermarking. Beyond simply detecting AI-generated text, the goal is to encode and recover a specific message $m \in \{0, 1\}^b$—such as a user identifier for personalized attribution. This enables answering not just "*is this AI-generated?*" but "*which specific user or system generated this text?*" Such fine-grained attribution is essential for large-scale deployment where accountability matters.

Existing watermarking methods struggle with these requirements. Token-level approaches like KGW [3] and EXP [4] require direct access to model logits, excluding API-based deployment, and can degrade text quality through probability manipulation. Syntactic methods [5] fail to generalize across languages, while specialized approaches [6] work well in narrow domains but break down when applied more broadly. Even recent black-box methods [7, 8] rely on surface-level statistics or require auxiliary models, limiting their robustness and scalability.

We introduce SAEMARK, a fundamentally different approach that sidesteps these limitations entirely. Our key insight is deceptively simple: different LLM generations exhibit distinct patterns in their semantic features, and these patterns can be leveraged for watermarking through *selection* rather than *modification*. Instead of altering how text is generated, we generate multiple candidates and choose those whose feature patterns align with a watermark key.

This approach works by operating on meaningful *units* of text—sentences for natural language, functions for code. For each unit, we extract deterministic features that capture semantic properties, compute a scalar statistic, and normalize it to behave predictably across different texts. Using the watermark key, we derive target values for each position. During generation, we sample multiple candidates from the LLM and select the one whose feature statistic is closest to the target, ensuring the final sequence encodes the desired message. The elegance lies in what we *don't* change: no model weights, no logit manipulation, no token modifications. Every selected text segment is a natural LLM output, preserving quality while enabling attribution. The approach works with any LLM through API calls, generalizes across languages and domains, and provides theoretical guarantees on watermark success that scale predictably with computational budget.

Our contributions span theory and practice. We develop a **general framework** for watermarking through feature-guided selection that works with any feature extractor and LLM. We provide **theoretical guarantees** that explain how SAEMark accuracy scales with compute budget, independent of feature extractors. Finally, we demonstrate a **practical instantiation** using SAE as feature extractor that achieves superior accuracy and text quality across languages and domains, encoding more information than existing approaches.

## 2   Related Work

**LLM watermarking** is a technique to embed special patterns into the output of LLMs, and has traditionally been used to identify LLM generated text from human-written text [9]. Different from *post-hoc detection* methods [10] that analyze statistical patterns in existing text, language model watermarking aims to embed detectable signatures during generation [3]. These methods *compromise generation quality* through direct manipulation of token probabilities [3] or syntactic modifications [11]. The challenge of *language and domain generalization* remains largely unaddressed, with current techniques primarily optimized for English and struggling with multilingual content or specialized domains like code [6]. Notably, PersonaMark [12] represents early attempts at personalized watermarking, but its reliance on English-specific syntactic patterns and closed-source implementation makes scalability and cross-lingual capability difficult to verify. Recently, more *multi-bit* watermarking methods have been proposed to embed multiple bits of information into gen-

erated text [13, 14, 15, 16, 17, 18], primarily by extending single-bit watermarking that manipulates logits during generation; these methods inherit the limitations of single-bit designs.

Complementary *black-box* watermarks avoid white-box logit access by using post-hoc selection or rewriting [8]. However, they typically operate on statistics or introduce auxiliary model dependencies and do not directly address multi-bit message embedding at scale. Our framework differs by performing inference-time *selection* among naturally generated candidates using *deterministic feature statistics*. This enables extractor-agnostic analysis and multilingual, domain-agnostic *multi-bit* watermarking without modifying model logits.

**Sparse Autoencoders (SAEs)** are pre-trained interpretability tools that decompose LLM activations into human-understandable features [19]. For a given base model $\mathcal{M}$ and layer $l$, an SAE processes hidden states $\mathbf{h}_t$ at position $t$ as:

$$\mathbf{f}_t = \text{SAE}_l(\mathbf{h}_t) \tag{1}$$

where $\mathbf{f}_t \in \mathbb{R}^m$ is a sparse vector (typically $m \gg \dim(\mathbf{h}_t)$) with $\leq 5\%$ active features. The SAE is trained through two objectives: 1) reconstruct original activations, and 2) enforce feature sparsity via $L_1$ regularization:

$$\mathcal{L} = \underbrace{\|\mathbf{h}_t - \text{Dec}(\mathbf{f}_t)\|^2}_{L_{rec}} + \lambda \underbrace{\|\mathbf{f}_t\|_1}_{L_{sparse}} \tag{2}$$

This training produces features that correspond to interpretable concepts [19, 20] like "Function definitions" or "Concept related to color blue" [21]. Our watermarking leverages key properties of pre-trained SAEs: **multilingual** activation allows the same features to fire for equivalent concepts across languages deterministically. **sparsity** enables efficient analysis through few active features per token. These properties support language-agnostic statistics via masked feature aggregations:

$$\phi(\mathbf{y}) = \frac{1}{|\mathbf{y}|} \sum_t \mathbf{f}_t \odot \mathbf{m} \tag{3}$$

where $\mathbf{m}$ filters background features that fire ubiquitously regardless of content (e.g., punctuation). The summary $\phi(\mathbf{y})$ provides a deterministic statistic used by our watermarking procedures (Sec 3).

## 3 Methodology

We present a general framework for post-hoc, multi-bit watermarking via feature-based rejection sampling. The key observation is that different LLM generations produce distinct values of *deterministic feature statistics* computed over domain-appropriate units, and these statistics can be steered by selecting among naturally generated candidates, without modifying model logits, parameters or generated texts. We structure the section as follows: (1) a general framework that is extractor-agnostic, (2) theoretical guarantees with an emphasis on worst-case bounds, and (3) an effective instantiation using sparse autoencoders as feature extractors.

### 3.1 Task Definition

We adopt a *multi-bit* view of attribution: beyond binary detection, the objective is to encode a message $m \in \{0,1\}^b$ that is recoverable at detection. Personalized attribution is a special case where $m$ encodes a user identifier bound to a key.

**Generation (multi-bit).** Given a base LLM $\mathcal{M}$, watermark key $k \in \mathcal{K}$, input $\mathbf{x}$, and message $m \in \{0,1\}^b$, the algorithm produces $\mathbf{y}$ by post-hoc selection over $\mathcal{M}$'s outputs (black-box/API-compatible; no access to logits or parameters; cf. black-box watermarking [7, 8]):

$$\mathbf{y} = \text{Mark}(\mathcal{M}, \mathbf{x}, k, m). \tag{4}$$

**Detection (multi-bit).** For any text $\mathbf{y}'$ and key $k$, the detection algorithm outputs a decoded message or reject:

$$\text{Detect}(k, \mathbf{y}') \to m \ \text{ or } \ \perp . \tag{5}$$

**Threat model and scope:** The scheme targets three properties: *key privacy* (deriving $k$ from watermarked outputs is hard), *verifier-held detectability* (any party holding $k$ can verify), and *collusion resistance* (multiple keys should not facilitate forgery). Our focus is attribution without storing LLM generated text. This work does not claim cryptographic unforgeability when keys are known; preventing adversarial forgeries is an important direction for security-focused follow-ups.

## 3.2 General Framework for Feature-based Watermarking

Our approach operates on a simple intuition: suppose we have a deterministic feature extractor that maps any text sequence into a scalar value, where such values follow a predictable distribution (e.g., approximately normal) for naturally generated text. Given a watermark key $k$ encoding multi-bit information, we can derive a sequence of target scalar values from this key. During generation, we produce text chunk by chunk, ensuring each chunk yields a scalar value with the smallest difference to its corresponding target—effectively implementing rejection sampling guided by our feature-based reward function. This process steers generation toward key-dependent patterns without modifying the underlying language model.

| **Algorithm 1:** Watermark generation | **Algorithm 2:** Watermark detection |
|---|---|
| **Input:** Prompt $c$, key $k$, LLM $G$, extractor $\phi$, statistic $s(\cdot)$ with CDF estimate $\hat{F}$, units $M$, attempts $K$, candidates $N$ 
 **Output:** Watermarked text $x^*$ 
 **for** $attempt \leftarrow 1$ **to** $K$ **do** 
    $x^* \leftarrow c$; $\{\tau_i\}_{i=1}^M \leftarrow$ TargetsFromKey$(k)$; 
    $\{z_i\} \leftarrow \emptyset$ 
    **for** $i \leftarrow 1$ **to** $M$ **do** 
      $\mathcal{X} \leftarrow$ GenerateCandidates$(G, x^*, N)$ 
      $x_{best} \leftarrow \arg\min_{x \in \mathcal{X}} \left\| \hat{F}(s(\phi(x))) - \tau_i \right\|$ 
      $x^* \leftarrow x^* \oplus x_{best}$; $z_i \leftarrow \hat{F}(s(\phi(x_{best})))$ 
    **end** 
    **if** CheckAlignment$(\{\tau_i\}, \{z_i\})$ **then** 
      **return** $x^*$ 
    **end** 
 **end** 
 **return** $x^*$ | **Input:** Text $x$, candidate keys $\mathcal{K}$, extractor $\phi$, statistic $s(\cdot)$ with CDF estimate $\hat{F}$, alignment thresholds, significance $\alpha$ 
 **Output:** Detection result $d \in \mathcal{K} \cup \{\emptyset\}$ 
 $\{z_j\} \leftarrow [\hat{F}(s(\phi(u))) \; \forall u \in$ SegmentByDomain$(x)]$ 
 $\mathcal{D} \leftarrow \emptyset$ 
 **foreach** $k_i \in \mathcal{K}$ **do** 
    $\{\tau_j\} \leftarrow$ TargetsFromKey$(k_i, |\{z_j\}|)$ 
    **if** CheckAlignment$(\{\tau_j\}, \{z_j\})$ **then** 
      $t, p \leftarrow$ StudentTTest$(\{z_j\}, \{\tau_j\})$ 
      **if** $t > t_{\alpha/2} \wedge p < \alpha$ **then** 
        $\mathcal{D} \leftarrow \mathcal{D} \cup \{(k_i, t)\}$ 
      **end** 
    **end** 
 **end** 
 **return** $\arg\max_{(k_i, t_i) \in \mathcal{D}} t_i$ if $\mathcal{D} \neq \emptyset$ else $\emptyset$ |

Figure 2: **Pseudocode for SAEMARK**: generation and detection.

**Text segmentation.** We segment text into smaller *units* $\{u_i\}_{i=1}^M$ such as sentences for natural language or function blocks for code. Each unit will carry one symbol of the watermark signal.

**Feature extraction.** A deterministic feature extractor $\phi : \mathcal{U} \to \mathbb{R}^d$ maps each text unit to a feature vector, from which we compute a *scalar statistic* $s(u) = g(\phi(u)) \in \mathbb{R}$. Crucially, we assume that this statistic follows a predictable distribution when computed over naturally generated text units.

**Statistical normalization.** To enable analysis independent of the specific feature extractor, we normalize the statistic $s(u)$ to a standard range $[0, 1]$ using its empirical distribution. Specifically, we estimate the cumulative distribution function $F_S$ from natural text, then map each unit's statistic via $z(u) = \hat{F}(s(u))$ where $\hat{F}$ is the empirical CDF. This ensures $z(u)$ values are approximately uniformly distributed for natural text.

**Watermark generation process.** Given a watermark key $k$ encoding multi-bit information, we first randomly generate a sequence of target values $\{\tau_i\}_{i=1}^M$ by seeding a PRNG generator with $k$ and sampling each $\tau_i$ from a suitable range deterministically. Then, for each position $i$, we generate $N$ candidate text units from the LLM and select the candidate $c^*$ whose normalized statistic $z(c^*)$ is closest to the target $\tau_i$.

**Watermark detection process.** To detect a watermark in input text, we segment it into units, compute the normalized statistic $z(u)$ for each unit, and compare the resulting sequence $\{z_i\}$ against target sequences derived from candidate keys. We apply a two-stage CheckAlignment process to verify sequence before statistical testing.

The CheckAlignment process employs two critical filters to ensure the observed sequence $\{z_i\}_{i=1}^M$ and expected target sequence $\{\tau_i\}_{i=1}^M$ are sufficiently similar:

**Range Similarity Filter:** This constraint ensures the dynamic ranges of observed and target sequences are similar:

$$R_{\min} < \frac{z_{\max} - z_{\min}}{\tau_{\max} - \tau_{\min}} < R_{\max} \tag{6}$$

where $z_{\max} = \max_i z_i$, $z_{\min} = \min_i z_i$, and similarly for $\tau$. We set $R_{\min} = 0.95$, $R_{\max} = 1.05$.

**Overlap Rate Filter:** This constraint ensures sufficient overlap between the value ranges of both sequences:

$$\frac{|\{i : \tau_i \in [z_{\min}, z_{\max}]\}|}{M} \geq O_{\min} \tag{7}$$

where $M$ denotes the number of textual units in the sequence and $O_{\min} = 0.95$ ensures that at least 95% of target values fall within the observed range. These two filters aim to eliminate spurious matches: the range similarity filter prevents matching sequences with fundamentally different statistical properties, while the overlap rate filter ensures meaningful correspondence between target and observed values. Only after passing both alignment checks do we apply Student's t-test for statistical significance. The key with the highest significance score is returned if it passes the threshold; otherwise, we classify the text as unwatermarked.

## 3.3 Theoretical Analysis and Guarantees

We provide theoretical guarantees on watermark embedding success that enable reliable detection by a conservative bound. For clarity, we present our analysis for a single textual unit and refer to our experiments for empirical validation of multi-unit performance with `CheckAlignment` process.

**Embedding success under Gaussian assumption.** Let target values $\tau$ be sampled from the feasible range $[\mu - 2\sigma, \mu + 2\sigma]$ where the feature statistic follows $S \sim \mathcal{N}(\mu, \sigma^2)$. Given $N$ candidate generations with feature statistics $S_1, S_2, \ldots, S_N$, we seek the probability of finding at least one candidate within relative tolerance $k$ of our target:

$$\mathbb{P}(\exists j : |S_j - \tau| \leq k\tau) \geq 1 - (1 - p_{\min})^N \tag{8}$$

**Worst-case analysis and bounds.** To derive conservative guarantees, consider the worst-case target $\tau = \mu + 2\sigma$ at the boundary of the feasible range. The single-candidate success probability becomes:

$$p_{\min} = \mathbb{P}((1-k)\tau \leq S_j \leq (1+k)\tau) = \Phi\left(\frac{(1+k)(\mu+2\sigma) - \mu}{\sigma}\right) - \Phi\left(\frac{(1-k)(\mu+2\sigma) - \mu}{\sigma}\right) \tag{9}$$

where $\Phi$ denotes the standard normal CDF and $p_{\min} = \Phi(2(1+k) + k\mu/\sigma) - \Phi(2(1-k) - k\mu/\sigma)$.

The fundamental insight is that observed feature statistics are tightly bounded to target values. Setting strict tolerance $k$ guarantees strong detection accuracy: embedding succeeds with high probability $1 - (1 - p_{\min})^N$ even with conservative parameters, while detection maintains precision because legitimate watermarks exhibit tight statistical binding that unwatermarked text cannot match. This framework provides exponential improvement with candidate count $N$, enabling principled compute-accuracy tradeoffs validated empirically across diverse tasks.

## 3.4 Sparse Autoencoder Instantiation

What concrete feature extractor should we use? We need statistics that are deterministic, semantically meaningful, and statistically regular. Sparse autoencoders—interpretability tools designed to understand language model internals—provide an ideal solution. They decompose language representations into interpretable semantic components (e.g. "technical writing," "math symbols") that exhibit distinctly different activation patterns across generations. By applying the SAE to a separate "anchor" model, our approach remains compatible with any target language model, including API services, while extracting the discriminative yet predictable statistics our framework requires.

**The Feature Concentration Score intuition.** Rather than using raw sparse autoencoder outputs, we compute a Feature Concentration Score (FCS) that captures a fundamental property of coherent text: semantic focus. The key insight is that well-formed text tends to concentrate its semantic activation

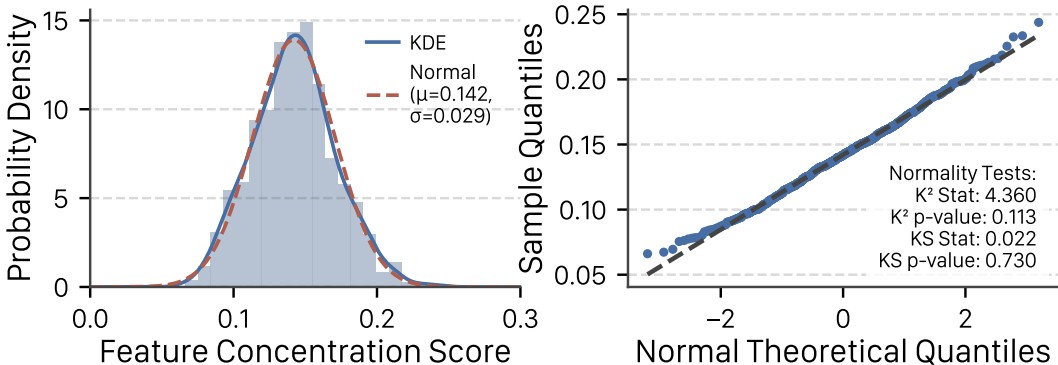

Figure 3: **Distribution analysis of FCS.** FCS distribution with density estimation (left) and Q-Q plot (right); statistical tests support approximate normality.

on a consistent set of relevant features, while unfocused or incoherent text spreads activation more uniformly. For example, a technical manual concentrates activation on features related to formal language and domain expertise, while creative writing focuses on narrative and stylistic features.

This concentration pattern provides an ideal watermark signal—we can steer generation toward specific concentration levels without affecting text quality, since both high and low concentration can correspond to natural, well-written text in different contexts. The FCS measures this by identifying the most salient feature activated by each token, then computing what fraction of the total activation mass concentrates in these important features:

$$\text{FCS}(T) = \frac{\sum_{t=1}^{n} \sum_{i \in S} \phi_{t,i}}{\sum_{t=1}^{n} \|\phi_t\|_1}, \tag{10}$$

where $S = \{\arg\max_i(\phi_{t,i} \odot m_i) : t = 1, \ldots, n\}$ contains the indices of the most salient features across all tokens, after applying the background mask $m$ and deduplication. This provides our framework's statistic $s(u) = \text{FCS}(u)$, which empirically follows approximately normal distributions across different domains and languages, validating our theoretical assumptions. We provide illustration and detailed analysis of this process in Appendix D.

**Implementation details.** Two practical optimizations bridge the gap between our theoretical framework and robust empirical performance. First, the CheckAlignment algorithm's eliminate spurious statistical matches that would otherwise compromise detection accuracy. Second, background feature masking ensures FCS calculations focus on discriminative semantic patterns rather than ubiquitous surface features. We precompute a mask excluding "background" SAE features, like those related to punctuation or basic grammar, to focus on discriminative semantic patterns. With empirically observed parameters $\mu = 0.142$, $\sigma = 0.029$, and tolerance $\epsilon = 0.1$, our bounds yield concrete success probabilities: $N = 50$ achieves $> 99\%$ per-unit success, $N = 20$ maintains $85.32\%$, and $N = 10$ achieves $61\%$. Since each generation involves multiple units, overall success rates significantly exceed these per-unit bounds. Modern inference engines support parallel generation of $N$ candidates simultaneously, making the approach practically efficient despite these extra compute overhead. We have extensive ablations and empirical results in the following experiments.

## 4 Experiments

Our experiments systematically address four fundamental questions: (1) How *accurate and quality-preserving* is our method compared to existing single-bit and multi-bit watermarks? (2) What are the *computational overhead characteristics and scalability properties* in practice? (3) How *robust* is our method against *adversarial attacks*? (4) Which *components* contribute most significantly to bridging the gap between theoretical bounds and empirical performance?

### 4.1 Experimental Setup

We evaluate on 4 diverse datasets as shown in Table 1. Following common practice in prior work, we report Accuracy, Recall, F1 at 1% FPR. For text quality, we report win-rates of pairwise comparison on PandaLM judged by GPT-4o in our main results, and average pointwise scores on BIGGen-Bench judged by their officially re-

Table 1: **Dataset Statistics.** Characteristics of the multilingual benchmarks used in evaluation.

|  | **C4** [22] | **LCSTS** [23] | **MBPP** [24] | **PandaLM** [25] |
|---|---|---|---|---|
| # Samples | 500 | 500 | 257[†] | 169 |
| Language | English | Chinese | Python | English |
| Task Type | Completion | Summarization | Code Generation | Instruction Following |

[†]From test split of sanitized version of MBPP.

leased judge model as an alternative text-quality experiment. We use implementation for single-bit watermarks from MarkLLM [26] toolkit and Waterfall [13] as it's the current best open-source training-free multi-bit watermark similar to our setting. Full details of baselines in Appendix C.

## 4.2 Watermarking Accuracy and Text Quality

Multi-bit watermarking poses a fundamentally harder challenge than single-bit detection: we must embed significantly more information into the same text length while maintaining both accuracy and quality. Despite this increased difficulty, Table 2 shows SAEMark achieves superior accuracy compared to both single-bit baselines and the current best multi-bit watermark across all domains.

**Accuracy across domains.** SAEMark establishes new state-of-the-art performance: **99.7% F1 on English**, **99.2% on Chinese**, and **66.3% on code**. Notably, we outperform specialized methods in their own domains—surpassing code-specific SWEET by **3.9 points F1** (66.3% vs. 62.4%) despite our general-purpose design. While other methods suffer severe cross-domain performance cliffs, SAEMark captures language-agnostic patterns that generalize across syntactic variations. The *multi-bit comparison* reveals particularly dramatic advantages: SAEMark outperforms the current best multi-bit method Waterfall by **6.5 points F1** on English (99.7% vs. 93.2%) and an exceptional **54.7 points** on code (66.3% vs. 11.6%), demonstrating semantic feature-based selection's clear superiority over vocabulary permutation approaches, especially in low-entropy domains.

**Text quality.** Beyond accuracy, Table 2 shows SAEMark achieves the **highest quality score (67.6%)** on PandaLM as judged by GPT-4o pairwise comparisons. To study how this generalizes across different backbone LLMs, we conduct additional evaluation on BIGGen-Bench comparing against watermarked baselines and unwatermarked text.

Table 3 confirms SAEMark *achieves the highest quality* among watermarks across three backbone LLMs. This quality advantage stems from our design: rather than manipulating logits or applying external rewriting to obtain watermarked text, we simply run *post-hoc selection* among naturally generated candidates, ensuring text quality stays bounded by the its own capabilities.

Table 2: **Comparison of Watermarks.** We generate watermarked and unwatermarked texts and then report detection performance at 1% false positive rate (FPR), all in single-bit settings. Best results are in **bold** and second-best are underlined. All metrics are reported as percentages (%).

| Method | C4 (English, [22]) | | | LCSTS (Chinese, [23]) | | | MBPP (Code, [24]) | | | PandaLM (Instruction, [25]) | | | |
|---|---|---|---|---|---|---|---|---|---|---|---|---|---|
| | Acc.↑ | Rec.↑ | F1↑ | Acc.↑ | Rec.↑ | F1↑ | Acc.↑ | Rec.↑ | F1↑ | Quality↑ | Acc.↑ | Rec.↑ | F1↑ |
| **Single-bit Watermarks** | | | | | | | | | | | | | |
| KGW [3] | 99.2 | 99.6 | 99.2 | 99.1 | 98.8 | 99.1 | 65.4 | 31.9 | 48.0 | 41.5 | **89.9** | **80.4** | **88.8** |
| EXP [4] | 99.5 | 99.6 | 99.5 | **99.3** | 99.4 | **99.3** | 57.8 | 16.7 | 28.4 | 23.2 | 79.3 | 59.4 | 74.2 |
| UPV [27] | 86.0 | 72.0 | 83.7 | 90.5 | 91.0 | 90.5 | 51.6 | 3.1 | 6.0 | 36.0 | 54.0 | 8.0 | 14.8 |
| Unigram [28] | 98.8 | 98.6 | 98.8 | 98.2 | 97.0 | 98.2 | 65.4 | 31.9 | 48.0 | 35.3 | 53.3 | 7.2 | 13.4 |
| DIP [29] | 96.0 | 92.6 | 95.9 | 97.7 | 96.2 | 97.7 | 60.7 | 22.6 | 36.5 | 36.5 | 81.5 | 63.8 | 77.5 |
| Unbiased [30] | 96.7 | 94.4 | 96.6 | 97.8 | 96.4 | 97.8 | 64.0 | 29.2 | 44.8 | 40.2 | 74.3 | 49.3 | 65.7 |
| SynthID [31] | 98.2 | 97.2 | 98.2 | 97.6 | 96.2 | 97.6 | 62.5 | 26.1 | 41.0 | 36.0 | 81.2 | 63.0 | 77.0 |
| SWEET [6] | 99.6 | 99.6 | 99.6 | 50.0 | 0.0 | 0.0 | 74.2 | 45.9 | 62.4 | 47.2 | 87.7 | 76.8 | 86.2 |
| **Multi-bit Watermarks** | | | | | | | | | | | | | |
| Waterfall [13] | 93.6 | 88.0 | 93.2 | 95.3 | 91.6 | 95.1 | 52.5 | 6.2 | 11.6 | 46.4 | 73.2 | 47.1 | 63.7 |
| **SAEMark (Ours)** | **99.7** | **99.8** | **99.7** | 99.2 | **99.6** | 99.2 | **74.5** | **50.2** | **66.3** | **67.6** | 86.6 | 73.9 | 84.6 |

Table 3: **Text quality evaluation on BIGGen-Bench.** Scores are on a 5-point Likert scale (higher is better) [32].

| Model | Unwatermarked | SAEMark | KGW | Waterfall |
|---|---|---|---|---|
| Qwen2.5-7B-Instruct | 4.13 | **4.05** | 3.97 | 4.02 |
| Llama-3.2-3B-Instruct | 3.69 | **3.85** | 3.56 | 3.62 |
| gemma-3-4b-it | 4.26 | **4.23** | 3.98 | 4.19 |

## 4.3 Computational Overhead and Scalability

Our theoretical analysis suggested requiring N=50 candidates to achieve 99%+ accuracy per unit. However, through the two practical optimizations in our framework: background feature masking and `CheckAlignment` filters, we achieve strong performance with significantly reduced computational overhead in practice.

(a) Perf. vs. Sampled Candidates

|  | N=5 | N=10 | N=20 | N=50 |
|---|---|---|---|---|
|  | | *C4* [22] | | |
| **Acc.** | 98.7 | 99.2 | 98.7 | 99.7 |
| **Rec.** | 77.4 | 96.8 | 98.7 | 99.8 |
| **F1** | 86.8 | 98.0 | 98.7 | 99.7 |
|  | | *LCSTS* [23] | | |
| **Acc.** | 98.6 | 99.0 | 98.6 | 99.2 |
| **Rec.** | 72.6 | 96.0 | 98.0 | 99.6 |
| **F1** | 83.6 | 97.5 | 98.6 | 99.2 |

(b) Perf. vs. End to end Latency

| Method | Acc. | Rec. | F1 | Latency |
|---|---|---|---|---|
| KGW | 99.0 | 99.5 | 98.9 | 3.24x |
| UPV | 90.3 | 86.3 | 89.5 | 2.35x |
| DIP | 99.5 | 99.7 | 99.5 | 3.29x |
| Waterfall | 98.8 | 97.3 | 98.1 | 1.06x |
| Ours(N=50) | 99.5 | 99.7 | 99.5 | 1.00x |

Figure 4: **Computational overhead analysis.** (a) Performance vs. number of sampled candidates for SAEMark. (b) Performance vs. avg latency across different watermarks.

**Practical efficiency.** Figure 4 (a) demonstrates this efficiency gain: **N=10 achieves 98.0% F1** on English with reasonable overhead, while even **N=5 attains 86.8% F1**—substantially better than our conservative theoretical bounds predicted. This flexibility enables deployment across different computational budgets. Moreover, subfigure (b) reveals a remarkable result: SAEMark achieves **99.5% F1 at 1.00× baseline latency**, substantially outperforming methods requiring **3.24× latency** (KGW) and **3.29× latency** (DIP) for comparable accuracy.

**Infrastructure advantage.** This performance difference reflects a *genuine architectural advantage*. Since SAEMark requires no logit manipulation, we can leverage highly optimized inference backends like TGI with parallel candidate generation and tricks like prefix caching and custom, optimized CUDA kernels. In contrast, these optimized frameworks do not provide efficient watermark implementations for logit-manipulation methods, as such implementations require significant backend rewriting and may impact performance. While this creates a significant latency difference despite some methods theoretically needing less compute overhead, we consider this a practical advantage reflecting the current state of inference infrastructure and these numbers reflect real deployment advantages of SAEMark.

**Multi-bit scaling.** Figure 6 shows our approach maintains over **90% accuracy up to 10 bits** (effectively differentiating 1,024 users) and **75% accuracy at 13 bits** (8,192 users), substantially exceeding Waterfall's performance through our high-dimensional SAE feature space. Importantly, this does not mean our method is only effective with 1,024 users—we are conducting fixed text length comparisons for fair evaluation. The superior information density stems from finer-grained semantic distinctions our framework enables.

## 4.4 Adversarial Robustness and Ablation Studies

**Adversarial Robustness.** Semantic SAE features provide inherent robustness against paraphrasing attacks. Figure 5 demonstrates our method's resilience across three attack types—word deletion, synonym substitution, and context-aware substitution. SAEMark shows strong resilience to such

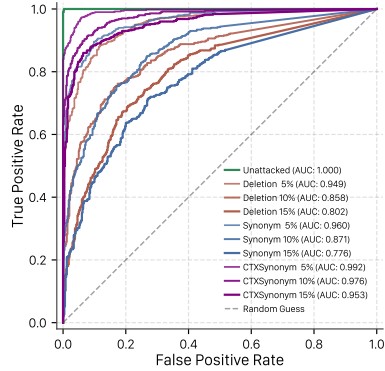

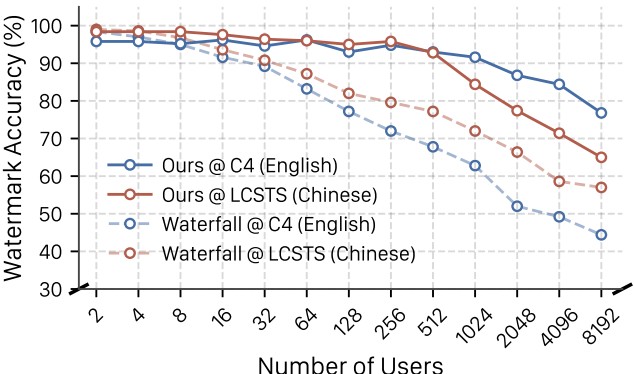

Figure 5: **Adversarial robustness.** ROC curves showing robust performance against three attack types with varying intensities.

Figure 6: **Multi-bit scaling and information density.** Watermark acc. across different message bits at fixed text length, demonstrating superior information density compared to multi-bit baselines with $\geq 90\%$ acc. up to 10 bits.

attacks. Due to space limitations, extended results testing attack intensities up to 50% are provided in Appendix E, demonstrating continued robustness even under stronger attacks.

**Ablation Studies** primarily involves validating the following components' contribution to SAEMark's empirical success: `CheckAlignment` **filters.** The Range Similarity and Overlap Rate filters prove *theoretically grounded and empirically validated.* Figure 7 (left) demonstrates that the `CheckAlignment` algorithm's 95% thresholds are not arbitrary—deviations cause significant degradation beyond 1,024 users (10 bits), confirming our theoretical analysis that these values optimally balance generation feasibility with discriminative power. These filters successfully compensate for theoretical independence assumptions when sequential generation creates dependencies in practice. Figure 7 (right) shows that the empirically-derived parameters achieve optimal ROC performance, validating our framework's theoretical foundations.

**Background feature masking.** This implementation detail proves *essential* for signal quality. Figure 12 in the appendix shows that removing the background mask causes **AUC to plummet from 1.0 to 0.85**. The mask excludes ubiquitous features like those related to punctuation or basic grammar patterns that would otherwise dominate FCS calculations without providing discriminative signals between different watermark keys. Detailed ablation results are provided in Appendix E.

These components work synergistically to enable SAEMark's practical success: background masking isolates meaningful signals while alignment constraints makes watermark detection more accurate than the theoretical settings.

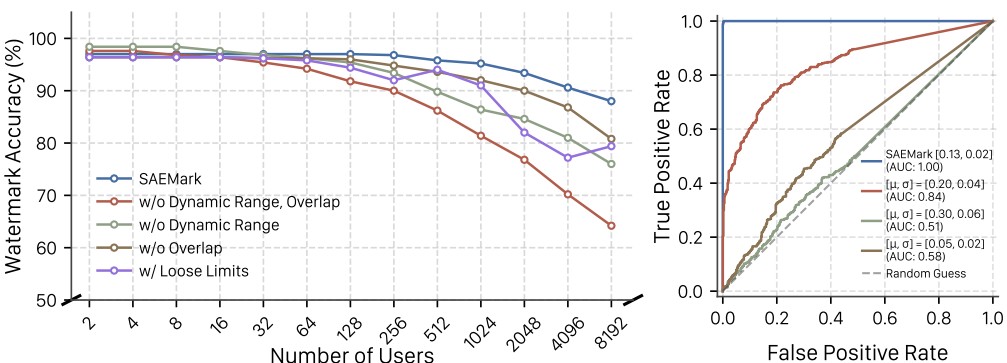

Figure 7: **Framework component ablation studies.** Left: Multi-bit watermarking accuracy scaling analysis with ablations. Right: ROC curves for feature concentration hyperparams ($\mu, \sigma$).

# 5  Conclusion

SAEMARK introduces a paradigm shift in AI-generated content attribution through feature-based rejection sampling. Our approach addresses critical limitations of existing watermarking methods by operating entirely through inference-time selection rather than model modification, enabling deployment with API-based services while maintaining superior text quality and detection accuracy. Three key advances ensures success: First, our general framework provides theoretical guarantees that relate watermark accuracy to computational budget, independent of the specific feature extractor used. Second, the sparse autoencoder captures meaningful semantic patterns that generalize across domains and languages and works great as a feature extractor. Third, practical optimizations bridge the gap between theoretical bounds and empirical performance, enabling efficient deployment. This work establishes that model interpretability tools can be effectively repurposed for content attribution tasks. The decoupling of watermarking from generation dynamics opens new possibilities for scalable, quality-preserving attribution systems that work seamlessly with existing language model APIs across diverse applications and languages.

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

# A    Limitations

We also found some limitations with our current approach. First, the method's effectiveness depends on SAE feature quality. But be noted that this does not affect the applicability of our algorithm on the base LLMs, since we only apply SAEs on the Anchor LLM and require only access to the output texts from the base LLM, and we have a lot of pretrained SAEs from the open-source community that exhibit strong performance in interpreting model outputs. Second, detection watermarks effectively requires open-ended generation tasks, making attribution challenging for very short outputs like multiple-choice problems that only contain option keys. However this is a universal challenge for all watermarking algorithms, since short texts inevitably contains less information and less space to inject additional signatures.

These constraints reflect tradeoffs in privacy-preserving watermarking. Future work could explore dynamic candidate pruning to address these limitations. Nevertheless, our experiments across 4 benchmarks suggest these constraints pose manageable practical impacts compared to the system's ethical advantages.

# B    Experimental Setup and Hyperparameter Details

This appendix provides a comprehensive description of the experimental setup, encompassing the hyperparameters and software configurations employed in this study.

## B.1    Hyperparameters (SAEMARK)

The following hyperparameters were used for the SAEMARK:

- **Candidate Number (N):** 50. This parameter denotes the number of candidate sequences sampled from the LLM.
- **Unit Number (M):** 10. This specifies the number of discrete generation *units* produced by the model per attempt.
- **Attempt Number (K):** 15. This metric represents the maximum times that the algorithm attempts to get an alignment.

## B.2    Model Configuration

The section outlines the hyperparameter by the model during generation.

- **Base Model:** `Qwen2.5-7B-Instruct`. This is the model on which the algorithm operates.
- **Sampling:** This algorithm enables the model to generate various candidates, for which the parameter $do\_sample$ is set to $True$.
- **Temperature:** This controls the randomness of the predictions by scaling the logits. The metric is set to $0.7$.
- **Max New Tokens:** This specifies the maximum number of new tokens that the model can generate, which is 20 during generation.

# C    Introduction to baselines

## C.1    Single-bit Watermarks

**KGW [3]**    The Key-based Green-list Watermarking (KGW) algorithm is a modern approach for watermarking text generated by LLMs. This method builds upon the work of [3], who introduced a watermarking scheme that divides the token set into 'red' and 'green' lists based on a secret key and previously generated tokens.

Key features of KGW include the bifurcation of the token set into 'red' and 'green' lists, the use of a random seed dependent on a secret key and hash of prior tokens, reweighting of token log-probabilities to favor green tokens, and the introduction of permutation-based reweight strategies.

These elements work in concert to create an effective watermarking system that balances detectability with output quality preservation.

The approach offers a balance between watermark embedding and preservation of text quality, addressing challenges faced by previous watermarking methods.

**Unigram [28]**   The Unigram-Watermark and KGW algorithms, both designed for watermarking LLM-generated text, have distinct characteristics. Unigram-Watermark operates on individual tokens, using a consistent green list for each new token, while KGW employs a $K$-gram approach with varying green lists. Unigram-Watermark's simplicity offers enhanced robustness against editing attacks and requires minimal implementation overhead. This streamlined approach leads to potential efficiency gains in both watermark embedding and detection processes, setting it apart from the more complex K-gram nature of KGW.

**SWEET [6]**   The Segment-Wise Entropy-based Embedding Technique (SWEET) is an innovative approach to watermarking code generated by large language models. SWEET addresses the challenge of maintaining code functionality while embedding detectable watermarks. It operates by selectively applying watermarking to high-entropy segments of the generated code, thereby preserving the overall code quality. This method significantly improves code quality preservation while outperforming baseline methods in detecting machine-generated code. SWEET achieves this by removing low-entropy segments during both the generation and detection of watermarks, effectively balancing the trade-off between detection capability and code quality degradation.

**UPV [27]**   The key feature of UPV is its use of separate neural networks for watermark generation and detection, addressing the limitation of shared key usage in previous methods. This separation allows for public verification without compromising the watermark's security. UPV employs shared token embedding parameters between the generation and detection networks, enabling efficient and accurate watermark detection. The algorithm embeds small watermark signals into the LLM's logits during generation, similar to existing methods, but uniquely conceals the watermarking details in the detection process. This approach ensures high detection accuracy while maintaining computational efficiency, and significantly increases the complexity of forging the watermark, thus enhancing its security in public detection scenarios.

**DIP [29]**   The Distribution-Preserving Watermarking (DIP) algorithm represents a significant advancement in watermarking techniques for large language models (LLMs). DIP's innovation is its ability to maintain the original token distribution of the LLM while embedding a watermark, addressing a critical limitation of previous methods. This distribution-preserving property is achieved through a novel permutation-based approach that reweights token probabilities without altering the overall distribution. DIP offers provable guarantees on distribution preservation, detectability, and resilience against text modifications. The algorithm employs a texture key generation mechanism that considers multiple previous tokens, enhancing its robustness. Notably, DIP maintains text quality comparable to the original LLM output, owing to its distribution-preserving nature.

**Unbiased [30]**   Unbiased watermarking and DIP watermarking are closely related concepts in the field of text watermarking for large language models (LLMs). Both approaches aim to embed watermarks while maintaining the original distribution of the LLM's output. The key distinction lies in their theoretical foundations and implementation. Unbiased watermarking ensures that the expectation of the watermarked distribution matches the original distribution, while DIP watermarking guarantees that the watermarked distribution is identical to the original for every input. In essence, unbiased watermarking can be viewed as a relaxed version of DIP watermarking. While unbiased watermarking allows for small deviations in individual instances, DIP watermarking maintains strict distribution preservation. This relationship highlights a spectrum of watermarking techniques, where unbiased methods offer a balance between practicality and distribution preservation, while DIP methods provide stronger theoretical guarantees at potentially higher computational costs.

**SynthID [31]**   SynthID is an advanced watermarking method for large language models (LLMs) that builds upon previous work in generative text watermarking. The key innovation of SynthID lies in its use of Tournament sampling, which provides superior detectability compared to existing methods. This approach offers rigorous and customizable non-distortion properties, allowing for

text quality preservation while maintaining effective watermarking. SynthID has been empirically validated, including through real user feedback from millions of chatbot interactions. Notably, the method introduces an algorithm to combine generative watermarking with speculative sampling, enabling efficient deployment in high-performance, large-scale production LLMs.

**EXP [4]**  EXP employs a pseudorandom function $f_s()$ with a secret seed $s$ known only to the model provider. Given previous tokens $w_1, ..., w_{t-1}$ and GPT's probability distribution $p_1, ..., p_K$ for the next token $w_t$, the algorithm generates real values $r_i \in [0, 1]$ using $f_s()$. EXP then selects the token $i$ that maximizes $r_i^{1/p_i}$. To detect the watermark, it calculates $\sum_{t=1}^{T} \ln \frac{1}{1-r_t^t}$ and compares it to a threshold. The scheme preserves the original token distribution while embedding a detectable watermark, with theoretical analysis showing distinct expected values for normal and watermarked text. The number of tokens required for reliable detection is $O(\frac{1}{\alpha^2} \log \frac{1}{\delta})$, where $\alpha$ is the average entropy per token and $\delta$ is the acceptable misclassification probability.

## C.2  Multi-bit Watermarks

### C.2.1  Baselines

**CTWL [86]**  CTWL is a framework designed to embed multi-bit customizable information into texts produced by large language models (LLMs). It allows watermarks to carry details such as model version, generation time, and user ID. CTWL provides a mathematical model for watermarking and a comprehensive evaluation system that considers factors like success rate, robustness, coding rate, efficiency, and text quality. The Balance-Marking method uses a proxy language model to partition vocabulary probabilities, aiming to maintain watermarked text quality and achieve strong performance in evaluations. CTWL seeks to integrate multi-bit information watermarks into LLMs and offers a practical approach for tracing machine-generated texts.

**Waterfall [13]**  Waterfall is a training-free framework designed for robust and scalable text watermarking. It leverages large language models (LLMs) as paraphrasers to generate diverse text variations while preserving semantic meaning. By combining vocab permutation with orthogonal perturbation techniques, Waterfall aims to achieve scalability and robust verifiability while maintaining text fidelity. The framework supports multi-bit watermarks, enabling it to accommodate multiple users while ensuring effective watermark detection. Waterfall allows for a trade-off between watermark strength and text quality, making it adaptable to various requirements.

**CODEIP [17]**  CODEIP embeds a multi-bit message into generated code by softly biasing token logits during decoding. At each step, a hash of the previous token and the secret ID selects "watermark" tokens whose logits $L_{WM}$ are boosted proportional to a strength $\beta$. To guarantee syntactic validity, a pretrained type predictor assigns logits $L_{TP}$ only to tokens matching the expected lexical category, scaled by $\gamma$. The final next-token logits combine the base LLM scores, watermark bias, and grammar bias via

$$w_i = \arg\max_{w \in V} \text{softmax}\big(L_{LLM} + \beta L_{WM} + \gamma L_{TP}\big)$$

Extraction recovers the ID by re-hashing and finding the message maximizing cumulative watermark contributions.

**REMARK-LLM [14]**  REMARK-LLM introduces a robust watermarking framework for texts generated by large language models. It consists of three modules: message encoding, reparameterization, and message decoding. The message encoding module uses a sequence-to-sequence model to embed watermarks into LLM-generated texts. The reparameterization module applies Gumbel-Softmax to the dense token distribution into a sparser form. The message decoding module extracts watermarks using a transformer-based decoder. The framework incorporates malicious transformations during training to enhance robustness against attacks.

**Provably Robust Multi-bit Watermarking for AI-generated Text [18]**  The authors propose a multi-bit watermarking scheme that embeds a user's bit-string ID into LLM-generated text by first partitioning the message into pseudo-randomly assigned bit-segments per token, then biasing "green

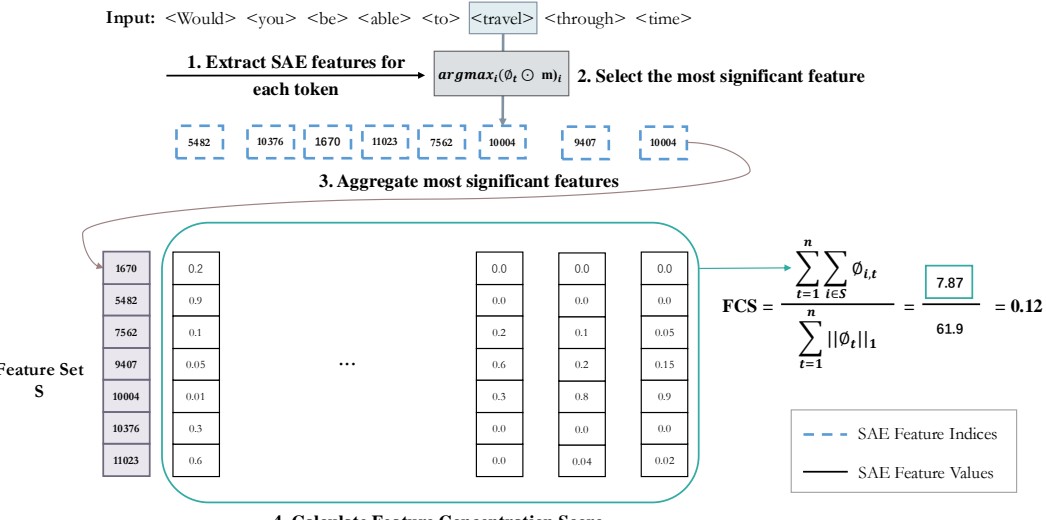

Figure 8: An example of Feature Concentration Score (FCS) calculation process.

list" logits seeded by each segment's value. A dynamic-programming step balances token-to-segment assignments, and a Reed–Solomon error-correction layer encodes segments to correct editing errors. Extraction enumerates only each segment's possibilities $O(k \cdot 2^{b/k})$, yielding efficient, provably robust watermark recovery under bounded edit distance.

**Robust Multi-bit Text Watermark with LLM-based Paraphrasers [87]** The authors propose a robust multi-bit text watermarking method using LLM-based paraphrasing. They design an encoder with two fine-tuned LLM paraphrasers ($\theta_0$ and $\theta_1$) that generate watermarked text by alternating based on the watermark code. A text segmentor divides the text into sentence-level segments, allowing each segment to encode one bit of the watermark message. The decoder uses a trained text classifier to determine the watermark bit for each segment. The method employs a co-training framework where the encoder and decoder are alternately updated. The decoder acts as a reward model during PPO-based reinforcement learning to fine-tune the encoder, optimizing both the detection of the watermark and the semantic similarity to the original text.

**Robust Multi-bit Natural Language Watermarking through Invariant Features [15]** The authors propose a robust multi-bit natural language watermarking method based on invariant features. They identify key words and syntactic dependencies as invariant features to embed watermarks, leveraging these features' resistance to minor textual modifications. A corruption-resistant infill model is also introduced to enhance watermark extraction robustness. Their method first selects mask positions based on these invariant features and then generates watermarked texts using an infill model. A robust infill model is developed to improve recovery of watermarked texts from corrupted versions.

### C.2.2 Why Choose Waterfall as the Multi-Bit Baseline?

Among the various text watermarking methods, Waterfall offers distinct advantages for multi-bit applications. As a training-free framework, it efficiently generates diverse text variations using LLMs as paraphrasers while preserving semantic meaning. Its key advantages include complexity that doesn't depend on word or sentence count, allowing scalability. It also offers evaluation metrics akin to single-bit methods and supports multi-language and multi-dataset watermarking, making it highly adaptable.

## D Details of FCS Generation

This section elaborates on the methodology behind the generation of the Feature Concentration Score (FCS). The process is illustrated in Figure Figure 8, which outlines four key steps.

**Algorithm 3:** ComputeFCS($\theta(T)$)

---

**Input:** Token sequence $T$, SAE $\theta$ for the entire sequence
**Output:** Feature Concentration Score (FCS)
$\Phi \leftarrow \theta(T)$, yielding activation vectors $\phi_1, \phi_2, ..., \phi_n$ for each token position in $T$;
$indices \leftarrow []$;
**for** $t = 1$ to $n$ **do**
    $\phi_t$ is the activation vector for token at position $t$;
    $index \leftarrow \arg\max_i(\phi_t \odot m)_i$;
    Append $index$ to $indices$;
**end**
$featureSet \leftarrow set(indices)$, removing duplicates;
$featureSum \leftarrow 0$;
$totalNorm \leftarrow 0$;
**for** $t = 1$ to $n$ **do**
    $tokenSignificance \leftarrow 0$;
    **foreach** $i \in featureSet$ **do**
        $tokenSignificance \leftarrow tokenSignificance + \phi_{t,i}$;
    **end**
    $featureSum \leftarrow featureSum + tokenSignificance$;
    $totalNorm \leftarrow totalNorm + ||\phi_t||_1$;
    // Accumulate significant features and norms
**end**
$FCS \leftarrow \frac{featureSum}{totalNorm}$;
// Calculate final FCS
**return** $FCS$;

---

**Extracting SAE Features for Each Token**    Given a token sequence $T$, we utilize SAE to derive an activation vector $\phi_t$ for each token position $t$. This vector, $\phi_t$, embodies the representation of the token at position $t$ with a dimensionality of 16,384.

**Selecting the Most Significant Feature**    For every activation vector $\phi_t$, our objective is to identify the most significant feature, which serves as a descriptor for the token at position $t$. This is achieved through applying the function $argmax_i(\phi_t \odot m)_i$, where $m$ is a mask. The output of this function yields the indices corresponding to the most prominent feature, denoted as "SAE Feature Indices" in Figure Figure 8.

**Aggregating Most Significant Features**    As depicted in Figure Figure 8, each token's position $t$ has its most significant feature. However, when summarizing the critical features of the entire sequence $T$, redundancies may occur. To address this, we employ a set operation to eliminate duplicate entries among the significant features, resulting in a unique collection termed as "Feature Set $S$".

**Calculating Feature Concentration Score**    Upon obtaining the Feature Set $S$, we aim to quantify how these significant features contribute to the overall sequence $T$ concerning SAE feature values. For each $\phi_t$, we compute the sum of $\phi_{t,i}$, where $i$ represents the index belonging to $S$. This aggregate score measures the contribution of significant features to individual tokens within $T$. Accumulating this metric across all tokens provides a global measure for the sequence.

To evaluate the total activation value of SAE features over the sequence $T$, we apply the $L1$ norm to each $\phi_t$, obtaining the sum of absolute values for each token's feature vector. Summing these across all tokens yields the total SAE value for $T$. The Feature Concentration Score (FCS) is defined as the ratio of the accumulated contributions of significant features to the total SAE feature values.

The detailed steps for computing the FCS are outlined in algorithm 3.

This score effectively captures the concentration of key features within a token sequence and is useful for applications in watermark embedding.

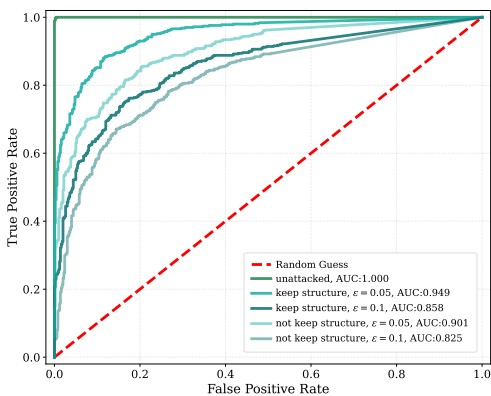

Figure 9: **Word Deletion on SAEMark** ROC curves highlighting the performance difference between "keep structure" and "not keep structure" methods under word deletion attacks with varying intensities (5%, 10%).

# E    Additional Experimental Results

## E.1    Adversarial Robustness Evaluation

**Word Deletion Attack**    In the main text, we conducted experiments using the "maintain content structure" version of the word deletion attack. However, the original word deletion attack involves splitting a paragraph and randomly removing words, which disrupts the structure that watermarking methods rely on, making it harder for the detection system to identify the watermark. To address this issue, we modified the attack to preserve structure while still performing word deletions. By maintaining the integrity of the structure, the attack bypasses watermark detection more effectively.

In our experimental results, we compare two versions of the word deletion attack. The "keep structure" method, represented in a darker color, shows more robust performance with higher AUC values (0.949 at $\epsilon = 0.05$ and 0.858 at $\epsilon = 0.1$). In contrast, the "not keep structure" method, shown in a lighter color, demonstrates a decline in performance, with AUC values dropping to 0.901 at $\epsilon = 0.05$ and 0.825 at $\epsilon = 0.1$. These results indicate that preserving the content structure during the attack strengthens the watermark's resistance, whereas random word deletions that disrupt the structure reduce detection accuracy.

As shown in the Figure 9, the "keep structure" method outperforms the "not keep structure" method in terms of AUC, demonstrating its effectiveness in watermark resistance.

**Basic Synonym Substitution Attack**    Our study also examines "keeping structure" versus "not keeping structure" approaches in the context of basic synonym substitution attacks, which are less likely to disrupt the content's structural integrity.

Figure 10 shows ROC curves comparing model performance under different conditions, with the original non-structure-preserving method in lighter shades and the modified structure-preserving method in darker hues. The analysis reveals minimal differences in AUC values between the two, indicating similar model resilience to both forms of synonym substitution. Notably, the model demonstrates performance robustness that exceeds that observed in deletion attack scenarios, reflected by AUC scores that remain close to the baseline.

**Context-aware Synonym Substitution Attack**    Due to our algorithm's prominent performance against context-aware synonym attack. More intensities (20%, 30%, 40%, 50%) are carried upon these kinds of attacking.

The results of the context-aware watermarking method, shown in  Figure 11 tested under this attack, demonstrate substantial robustness. Even with high substitution ratios—up to 50% token replacement—the AUC remains relatively high, highlighting the method's ability to maintain detection performance under significant adversarial pressure. The ROC curves further corroborate this, showing that the true positive rate remains consistently high across varying false positive levels, even as attack

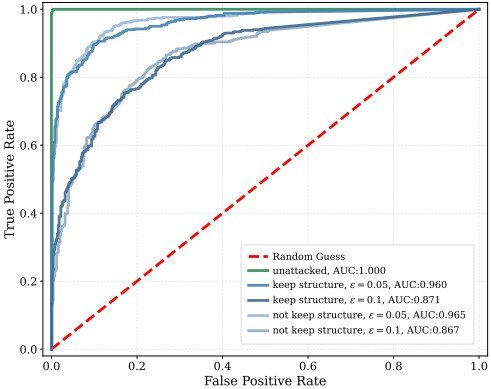

Figure 10: **Basic Synonym Substitution on SAEMark** ROC curves comparing "keep structure" and "not keep structure" methods under basic synonym substitution attacks at different intensities (5%, 10%).

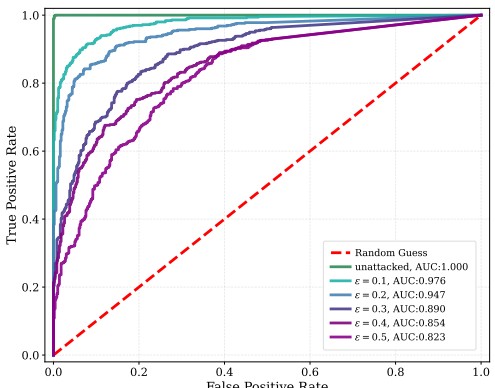

Figure 11: **Context-Aware Synonym Substitution on SAEMark** ROC curves comparing "keep structure" and "not keep structure" methods under basic synonym substitution attacks at different intensities (5%, 10%).

intensity increases. This demonstrates a well-balanced trade-off between true and false positives, ensuring reliable detection without excessive false alarms. These findings affirm that the watermarking method is both effective and robust, offering reliable protection against sophisticated attacks while maintaining strong detection accuracy.

### E.2  Ablation Study on Background Frequent Features

In section 3, we utilize $\phi_t \odot m$, where $m$ is a mask that excludes background frequent features.

In this section, we generate the Feature Concentration Score (FCS) without using $m$ and conduct ROC experiments for further analysis. To evaluate the impact of background frequent feature masking on our model's performance, we performed an ablation study.

With background frequent feature masking in place, the model achieved an AUC of almost 1.0. Upon removing this masking, the AUC dropped to 0.85, as illustrated in Figure 12. This significant decrease demonstrates that background frequent feature masking plays a crucial role in our algorithm, emphasizing its importance for optimal performance.

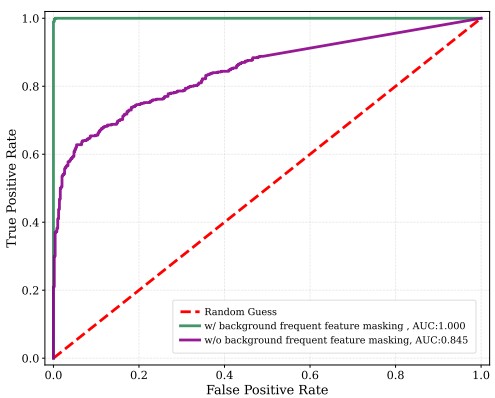

Figure 12: **Ablation on Background Frequent Feature Masking** The ROC curve compares the performance with and without background frequent feature masking.

# F    Use Of AI Assistants

We employed AI assistants for two tasks: (1) generating routine code implementations and boilerplate functions, and (2) performing grammatical review and sentence-level editing of the manuscript. All AI-generated content underwent thorough manual review. The core research methodology, findings, and analysis remain entirely our own work.

