# OpenReview forum: "SAEMark: Steering Personalized Multilingual LLM Watermarks with Sparse Autoencoders"
_NeurIPS.cc/2025/Conference — NeurIPS 2025 poster_

### Official Review · Reviewer_SdYH · 2025-06-07

**Clarity:** 2
**Significance:** 1
**Originality:** 2
**Rating:** 2
**Confidence:** 5

**Summary:**

This paper introduces SAEMARK, a watermarking method for LLMs that embeds personalized watermarks without requiring access to model logits or parameters. The key innovation is using Sparse Autoencoders to extract semantic features from generated text and then applying rejection sampling to select outputs whose feature concentrations match watermark-specific target distributions derived from user keys.

**Questions:**

- Could you clarify the distinction between your claimed ``multi-bit watermarking'' and what appears to be single-bit detection followed by user identification? How do you justify calling this multi-bit when no actual multi-bit information is extracted from individual watermarked texts?

- Given that your method supports only 1,200+ users, how do you envision scaling this approach to real-world deployment scenarios requiring millions of users? What are the theoretical and practical limits of your user identification approach?

- Your code generation results show only 66.3% F1 on MBPP, which is marginally better than traditional methods. Why do you claim effectiveness on code generation when the performance appears significantly inferior to your natural language results?

- Given that SAE features are likely to change substantially under semantic-preserving paraphrasing, how confident are you that your approach maintains robustness against such attacks? Can you provide theoretical or empirical justification for this robustness?

**Ethical Concerns:**

["NO or VERY MINOR ethics concerns only"]

**Limitations:**

Please refer to the weaknesses.

**Paper Formatting Concerns:**

-

**Quality:**

1

**Strengths And Weaknesses:**

Strength:
Enabling watermark injection without requiring access to model logits, which is a long-discussed issue.

Weaknesses:
1. The paper claims to be a ''multi-bit watermarking'' method, but this is fundamentally misleading. The approach only supports 1,200+ users, which is not a particularly large scale for real-world deployment. More importantly, the method does not actually extract multi-bit information from the watermarked text. Instead, it performs user matching by comparing observed feature patterns against different user-specific targets. This is essentially a single-bit detection problem (watermarked vs. non-watermarked) followed by user identification, rather than true multi-bit information extraction. The authors conflate user attribution with multi-bit watermarking, which are conceptually different tasks.

2. While the authors emphasize multilingual capabilities, many existing baselines (e.g., KGW) already support multiple languages effectively. The paper fails to clearly articulate what specific limitations existing methods have in multilingual settings that SAEMARK addresses. Furthermore, although the authors claim effectiveness on code generation, the experimental results show very limited performance on programming languages (66.3% F1 on MBPP), which is barely better than some traditional methods and significantly worse than their performance on natural language tasks.

3. The robustness experiments are concerningly limited, testing only up to 15% word deletion and synonym substitution attacks. Critically, the paper completely omits evaluation against paraphrasing and rewriting attacks, which are among the most practical and effective ways to remove watermarks. Many existing watermarking methods have specifically designed defenses against such attacks. The absence of these evaluations raises suspicions about whether the SAE-based approach is fundamentally vulnerable to content rewriting, since SAE features are likely to change significantly when sentences are paraphrased or rewritten, even if semantic meaning is preserved.

4. The core contribution appears to be simply applying existing SAE technology to watermarking, which represents limited technical innovation. The method seems relatively crude - essentially computing feature concentration scores and using rejection sampling to match target distributions. The approach lacks theoretical depth and appears more like an engineering application of existing interpretability tools rather than a fundamental advancement in watermarking techniques. The reliance on external SAE models also introduces additional dependencies and potential points of failure that are not thoroughly analyzed.

5. The experimental evaluation lacks comprehensive comparison with state-of-the-art robust watermarking methods, particularly those designed to withstand semantic-preserving attacks. The paper's focus on detection accuracy metrics while avoiding the most challenging attack scenarios suggests the evaluation may be cherry-picked to favor the proposed approach.

---

> ### Author Rebuttal · Authors · 2025-07-31
>
> Thank you for your review. While we see you have several significant concerns, we believe they may stem from a few *key misunderstandings and factual errors* regarding our method's contributions, novelty, and the experiments. We appreciate this opportunity to clarify these points and present new experimental results that we are confident will strengthen your evaluation of our work.
>
> **W1, Q1, Q2: Misunderstanding on Multi-bit Watermarking and Scalability**
>
> You pointed out that our method is "fundamentally misleading" by conflating user attribution with multi-bit watermarking and that supporting 1,200+ users is insufficient. **This reflects a core misunderstanding: Embedding a unique identifier for one of N users into watermarked text is, by definition, equivalent to embedding $log₂(N)$ bits of information into the text. This is the central challenge of multi-bit watermarking.**
>
> Our experiments, conducted under a **controlled text length**, show that SAEMark reliably injects **10+ bits** of information (corresponding to 1,024+ users with >90% accuracy ). **This is already state-of-the-art performance for a training-free multi-bit watermark**, significantly outperforming existing methods like Waterfall (which is a recent paraphrase-based multi-bit SoTA watermark), as shown in Figure 4 in our paper. The number of users we support scales directly with the length of the generated text, a universal property of all watermarking schemes. Clearly it is not possible to inject infinite extra information into texts of limited length.
>
>
>
> We add new experiments showing that when controlling for the **same total token generation budget**, which is the actual performance bottleneck, SAEMark is not only accurate (>99% F1) but also significantly faster than leading methods.
>
> | **C4**        | Acc.     | Rec.     | F1       | Latency |
> | ------------- | -------- | -------- | -------- | ------- |
> | KGW           | 98.8     | 99.2     | 98.8     | 174s    |
> | UPV           | 86.1     | 74.2     | 84.2     | 94s     |
> | DIP           | 99.3     | **99.8** | 99.3     | 85s     |
> | Waterfall     | 98.7     | 97.0     | 97.9     | 52s     |
> | SAEMark(N=50) | **99.7** | **99.8** | **99.7** | **40s** |
>
>
>
>
> | **LCSTS**     | Acc      | Rec      | F1       | Latency |
> | ------------- | -------- | -------- | -------- | ------- |
> | KGW           | 99.2     | 99.8     | 99.0     | 85s     |
> | UPV           | 94.5     | 98.4     | 94.7     | 94s     |
> | DIP           | **99.6** | **99.6** | **99.6** | 178s    |
> | Waterfall     | 98.8     | 97.6     | 98.2     | **33s** |
> | SAEMark(N=50) | 99.2     | **99.6** | 99.2     | 40s     |
>
>
>
> Additionally, although in our theoretical analysis we could achieve 99%+ accuracy with N=50 per unit, our empirical results yield strong performance at N=10 already. **Using only 20% of compute overhead is sufficient to produce empirically strong performance**, this is because typical scenarios include multiple units instead of only one (typical LLM calls would generate multiple sentences). We now report single-bit watermarking performance with SAEMark given different sets of N (which is the most significant factor for compute cost).
>
> | C4             | Acc  | Rec  | F1   | AUROC |
> | -------------- | ---- | ---- | ---- | ----- |
> | SAEMark (N=5)  | 98.7 | 77.4 | 86.8 | 0.99  |
> | SAEMark (N=10) | 99.2 | 96.8 | 98.0 | 0.99  |
> | SAEMark (N=20) | 98.7 | 98.7 | 98.7 | 0.99  |
> | SAEMark (N=50) | 99.7 | 99.8 | 99.7 | 1.0   |
>
> | LCSTS          | Acc  | Rec  | F1   | AUROC |
> | -------------- | ---- | ---- | ---- | ----- |
> | SAEMark (N=5)  | 98.6 | 72.6 | 83.6 | 0.98  |
> | SAEMark (N=10) | 99.0 | 96.0 | 97.5 | 0.99  |
> | SAEMark (N=20) | 98.6 | 98.0 | 98.6 | 0.99  |
> | SAEMark (N=50) | 99.2 | 99.6 | 99.2 | 1.0   |
>
> **W2, Q3: Multilingual and Code Generation Effectiveness**
>
> You claim our multilingual capabilities are not novel and that our code generation performance is "very limited." This is incorrect. Our method’s multilingual strength lies not in a specific component but in its complete **absence of language-specific design**. Unlike methods like KGW (which requires language-specific selection of red-list or green-list tokens) or paraphrase-based watermarks that require language-specific paraphraser or parsing modules, our approach is truly domain-agnostic.
>
> Take KGW as an example: KGW randomly selects green-list tokens based on a seed, however since modern LLMs are multilingual, if the user wants to inject watermarks into English texts but without any constraints, it would be likely some tokens from another language are chosen and would impact quality. In contrast we do not need such manual intervention, and this is why we achieve SOTA performance on both English and Chinese without any modification.
>
> Regarding code generation, it is a known challenging, low-entropy domain where all watermarking methods struggle. Acknowledging this difficulty is key. Our 74.5% accuracy (66.3% F1) on MBPP is not "barely better"; it is **state-of-the-art performance** that significantly outperforms all baselines, including specialized code watermarkers like SWEET (`Who Wrote this Code? Watermarking for Code Generation`, 72.4% accuracy, 45.9% F1).
>
> To further support our quality claims, we followed the suggestion of Reviewer R51M and ran new tests on the BIGGen-Bench, a multilingual text quality benchmark, which confirm SAEMark preserves quality best.
>
> | Model    | Unwatermarked | SAEMark (Ours) | KGW  | Waterfall |
> | -- | ------ | ---- | ---- | --------- |
> | `Qwen/Qwen2.5-7B-Instruct`   | **4.13**      | 4.05           | 3.97 | 4.02      |
> | `meta-llama/Llama-3.2-3B-Instruct` | 3.69          | **3.85**       | 3.56 | 3.62      |
> | `google/gemma-3-4b-it`             | **4.26**      | 4.23           | 3.98 | 4.19      |
>
> *(Scores on 5-point Likert scale, higher is better, Judged by the officially released model from BIGGen-Bench)*
>
> SAEMark consistently achieves the highest quality among watermarking methods, with minimal degradation from unwatermarked text. Additionally, when controlling for equal token generation budgets, SAEMark achieves the highest detection accuracy, demonstrating superior performance-per-compute.
>
>
>
> **W3, Q4: Robustness Against Paraphrasing and Rewriting Attacks**
>
> Thank you for emphasizing the critical importance of robustness, however there's an important misunderstanding: To improve readability, we report attack ratios ranging from 5% to 15% of 3 different attacks in Figure 5 of our paper and it already include 10 distinct ROC curves. **We actually have a dedicated section Appendix E, regarding adversarial robustness evaluation and Figure 9 to 11 shows attack ratios from 10% to 50% and our method still exhibit strong performance**, even in extreme scenarios like CTXSubstitution with $\epsilon=0.5$.
>
> The original paper already demonstrated strong resilience. We now add an extensive new experiments against sentence-level paraphrasing and extended our context-aware substitution attacks. The results demonstrate strong robustness. Similar to current paraphrasing watermark settings, we use a LLM paraphraser to randomly paraphrase parts of the text.
>
> | Type                | 10%  | 20%  | 30%  | 40%  | 50%  |
> | ------------------- | ---- | ---- | ---- | ---- | ---- |
> | **CTXSubstitution** | 0.98 | 0.95 | 0.89 | 0.85 | 0.82 |
> | **Sent Paraphrase** | 0.99 | 0.96 | 0.92 | 0.86 | 0.80 |
>
> Even when **50% of the sentences are paraphrased**, our method maintains a high AUC of 0.80. At a 50% context-aware substitution rate, the AUC is still 0.82. This directly proves that our method, which relies on deeper semantic features captured by SAEs, is fundamentally resilient to strong attacks.
>
>
>
> **W4: Novelty and Technical Contribution**
>
> You characterized our contribution as "simply applying existing SAE technology." We see how our paper might have given that impression, and we were perhaps too modest in presentation.  Our core innovation is a **fundamental paradigm shift in watermarking**: we propose **a general, training-free, logit-free framework based on rejection sampling**. Our theoretical analysis in Section 3.2 is novel, providing a clear and generally applicable foundation. **SAEs are merely one powerful instantiation of this framework. You could replace the SAE with *any* deterministic feature extractor whose features vary during generation and follow a normal-like distribution, and our theoretical guarantees would still hold.** This establishes a new, flexible, and powerful direction for post-hoc watermarking.
>
>
>
> **W5: Selection of baselines**
>
> We take your concern about "cherry-picking" baselines very seriously and want to clarify our evaluation process. We have a dedicated section on our selection of baseline methods in **Appendix C (from page 15 to 18), including introduction to every single baseline**.
>
> To ensure fairness and completeness, we used the standard **MarkLLM toolkit** , which implements numerous SOTA watermarking methods, and we reported results against all of them in Table 2. For our multi-bit experiments, we selected Waterfall as the primary baseline for clear and principled reasons: it is **(1) the current state-of-the-art multi-bit watermark , (2) open-source , and (3) training-free**, making it the most direct and fair competitor to our approach. **There was no cherry-picking.** **If there's any specific open-source watermarking method that you believe we have missed, we are happy to add new experiments during discussion period.**
>
> We trust these clarifications and the substantial new experimental evidence have thoroughly addressed your concerns. **SAEMark achieves what no previous method has: a training-free, logit-free, multi-bit, and robustly generalizable watermarking solution.**
>
> **We believe this is a significant and practical contribution to the field and respectfully ask you to reconsider your score in light of this new information.**

---

> ### Author Response · Authors · 2025-08-07
> **We'd love to engage in discussion**
>
> Dear Reviewer SdYH,
>
> Thank you for your review of our work SAEMark. As the rebuttal discussion period approaches its end (August 8th AoE), we wanted to reach out once more for a constructive discussion.
>
> We've provided comprehensive responses to each of your concerns, including extensive new experiments and clarifications that we believe address the issues you raised. These clarify fundamental points about multi-bit watermarking and demonstrate our unique achievement of training-free, logit-free watermarking that works across languages and closed-source APIs.
>
> We remain eager to discuss any remaining concerns and would greatly value your engagement. Thank you for your dedication to maintaining high standards in our research community.

---

### Official Review · Reviewer_R51M · 2025-06-30

**Clarity:** 3
**Significance:** 3
**Originality:** 4
**Rating:** 4
**Confidence:** 4

**Summary:**

This paper presents a new algorithm for text watermarking using sparse autoencoders. The key insight in the work is that different LLM samples to the same prompt exhibit vastly different features in a sparse autoencoder's space (SAE). Hence, via rejection sampling, an LLM sample can be chosen which closely matches the SAE statistic pre-chosen by a private key.

The authors perform several experiments on english/chinese/code output LLM detection and notice the proposed method outperforms several single and multi-bit watermarking baselines.

**Questions:**

Where exactly is the multilingual inductive bias in the method?

What is the exact difference between single and multi-bit watermarks? Would be good to highlight this early in the paper, it wasn't super clear to me.

**Ethical Concerns:**

["NO or VERY MINOR ethics concerns only"]

**Final Justification:**

Thanks to the authors for the detailed rebuttal and extra experiments. I will continue to support acceptance for the paper, but will not be able to raise my score further since my concerns about rejection sampling  (also voiced by other reviewers) + complexity remain.

Also I appreciate the BigGen experiments and they should be added to the paper. Evaluation on MATH/coding benchmarks will be very interesting -- these are low entropy tasks where there could be clear losses with this kind of repeated sampling.

**Limitations:**

Yes

**Quality:**

2

**Strengths And Weaknesses:**

*Strengths*

- The paper studies a research topic of growing importance every since the introduction of ChatGPT --- automatic watermarking algorithms for detecting AI-generated text.

- Unlike many previous watermarking methods, this method is post-hoc and does not modify the LLM generation process / logits. The rejection sampling ensures that all samples chosen were unperturbed LLM samples, ensuring high quality.

- The paper is generally well presented and easy to read.

- The proposed method outperforms several baseline watermarking algorithms in detection accuracy, while being better at preserving text quality (as judged by GPT-4o). The authors also include interesting experiments studying the robustness of their method to paraphrasing, ablations, and user-specific attribution.

*Weaknesses*

- Rejection sampling may not be suitable in all circumstances. For example, in lower entropy tasks (like factuality prompts), I would imagine the set of samples being produced across attempts are quite similar. How similar are the SAE / FCS features in this case? Does minor rewording the samples lead to significant changes in SAE space? Also, rejection sampling has its disadvantages. This will not work if the users set a low temperature in the models (which is common for some tasks), and it will kill any kind of LLM sample diversity for the user (which a user often desires).

- A lot more work should be done on the text quality side of experiments. The paper currently only includes one experiment on text quality (PandaLM), which is not commonly used by modern LLMs in 2025 releases. Perhaps quality analysis could be done with/without watermarking on the BigGen Bench which has a wide variety of tasks (https://arxiv.org/abs/2406.05761)? Even performance differences on reasoning benchmarks like MATH-500 or MMLU will be very helpful to see before/after watermarking.

- Minor, but the proposed method is quite complex. Is there no simpler method to derive discriminative features from SAE vectors, rather than deriving feature concentration scores? FCS scores do seem a critical part of the paper and should be properly described in the main body in my opinion. I'm especially confused about how exactly the target FCS determined from a random seed public key, and how do you ensure it's within the range of FCS across LLM samples?

Minor, but consider comparing against another post-hoc watermarking method introduced last year: https://arxiv.org/abs/2406.14517

---

> ### Author Rebuttal · Authors · 2025-07-31
>
> Thank you for your thoughtful and constructive review! We greatly appreciate your recognition of our work's originality and significance. Your positive assessment of our post-hoc approach and detailed feedback helps us strengthen the paper significantly.
>
>
>
> **Q1: Where exactly is the multilingual inductive bias in the method?**
>
> The multilingual capability in our method comes from the fact that we don't have any language-specific components or inductive biases—it's actually the *absence* of language-specific design that enables universal operation. Unlike previous methods that require language-specific components (KGW needs language-specific tokenizers, SWEET uses English syntax patterns, semantic watermarks need language-specific paraphrasers), **we only define unit boundaries (sentences for text, functions for code)**. **Everything else—feature extraction, FCS calculation, rejection sampling—operates identically across languages**. This domain-agnostic design is why we achieve 99.7% accuracy on English, 99.2% on Chinese, and 74.5% on code without any language-specific tuning.
>
>
>
> **Q2: What is the exact difference between single and multi-bit watermarks?**
>
> Thank you for highlighting this—we'll clarify this distinction upfront. Single-bit watermarking embeds one bit of information, answering only "is this AI-generated?" Multi-bit watermarking embeds multiple bits to encode additional information—in our case, which specific user among thousands generated the text. This is crucial for accountability: single-bit methods detect but cannot attribute. *Multi-bit watermarking is significantly harder as we must preserve multiple bits through generation while maintaining quality*. Our reliable multi-bit watermarking represents a major advance in making LLM outputs truly accountable.
>
> Although injecting 10 bits of information reliably seems modest, it is already state-of-the-art performance and this is indeed a challenging field.
>
>
> **W1: Rejection sampling limitations in low-entropy tasks**
>
> You raise an excellent point. For low-entropy tasks, we acknowledge the challenges. We now include additional analysis on TruthfulQA, where we collect responses and analyze the FCS score distribution, results show that factual QA responses exhibit lower FCS variance (0.017 vs 0.029) but still following a normal distribution, making watermark embedding harder but our theoretical framework still applies. This explains the performance degradation on constrained tasks like code generation (66.3% on MBPP).
>
> For deterministic settings (temperature=0), we can control the random seed based on the watermark key, making different keys produce different deterministic outputs. This maintains both reproducibility and watermark capability without requiring temperature sampling. We'll clarify these limitations and solutions in the revision.
>
>
>
> **W2: More comprehensive text quality evaluation**
>
> We completely agree and thank you for the BigGen-Bench suggestion! We conducted extensive new experiments across three models to comprehensively evaluate quality preservation:
>
> | Model                 | Unwatermarked | SAEMark (Ours) | KGW  | Waterfall |
> | --------------------- | ------------- | -------------- | ---- | --------- |
> | `Qwen/Qwen2.5-7B-Instruct`   | **4.13**      | 4.05           | 3.97 | 4.02      |
> | `meta-llama/Llama-3.2-3B-Instruct` | 3.69          | **3.85**       | 3.56 | 3.62      |
> | `google/gemma-3-4b-it`             | **4.26**      | 4.23           | 3.98 | 4.19      |
>
> *(Scores on 5-point Likert scale, higher is better, Judged by the officially released model from BIGGen-Bench)*
>
> SAEMark consistently achieves the highest quality among watermarking methods, with minimal degradation from unwatermarked text. Additionally, when controlling for equal token generation budgets, SAEMark achieves the highest detection accuracy, demonstrating superior performance-per-compute.
>
>
>
> **W3: FCS complexity and method description**
>
> You're absolutely right about needing clearer explanation. The target FCS for each segment is deterministically derived using the public key as a seed to sample from the observed distribution N(0.142, 0.029²), constrained to [μ-2σ, μ+2σ] for feasibility.
>
> More importantly, as you intuited, FCS is just one instantiation of our general framework. Any deterministic feature with natural variation could work. The core innovation is the rejection sampling framework for post-hoc watermarking, not the specific feature choice. We'll move FCS details to the main text and emphasize this generality.
>
> **Additional comparison**
>
> Thank you for suggesting the 2024 post-hoc work. This work is actually a simpler variant of Waterfall as both require a LLM paraphraser to inject watermark information by paraphrasing. We'll include a detailed comparison, noting that while it requires modifying the sampling process, our approach operates on completely unmodified LLM outputs.
>
>
> We're grateful for your constructive feedback and recognition of our originality. The BigGen-Bench results confirm that our method achieves superior performance while genuinely preserving output qualities. Combined with our multi-bit attribution capability and domain-agnostic design, we believe these improvements address your concerns comprehensively.
>
> Would these additions—especially the extensive quality evaluation and clearer explanations—be sufficient to demonstrate that our novel approach deserves strong support? We're committed to making this contribution as clear and well-evaluated as possible.

---

> > ### Comment · Reviewer_R51M · 2025-08-05
> > **Thank you for the rebuttal, will stick with "weak accept"**
> >
> > Thanks to the authors for the detailed rebuttal and extra experiments. I will continue to support acceptance for the paper, but will not be able to raise my score further since my concerns about rejection sampling  (also voiced by other reviewers) + complexity remain.
> >
> > Also I appreciate the BigGen experiments and they should be added to the paper. Evaluation on MATH/coding benchmarks will be very interesting -- these are low entropy tasks where there could be clear losses with this kind of repeated sampling.

---

### Official Review · Reviewer_wvir · 2025-07-01

**Clarity:** 3
**Significance:** 2
**Originality:** 3
**Rating:** 5
**Confidence:** 4

**Summary:**

This work introduces a novel technique for watermarking the outputs of black box LLMs based on extracting sparse codes from text chunks. They use techniques originally developed in the interpretability literature and combine a feature extractor (Anchor) LLM and a SAE model applied to its features to implement a multi-key(bit) watermarking scheme for attributing specific generations to specific keys (users). They empirically validate the performance of their method against existing LLM watermarking techniques and demonstrate an advantage in multi-key capacity, and detectability in domains like non-english text and code without impacting generation quality.

**Questions:**

1. Sec 3.1 Whats the JumpReLU "jump" hparam? and how is it tuned in this work?

2. What are the meanings of k_priv and k_pub? It appears that kpub is used at both generation and detection time, i.e. this is not actually an asymmetric scheme. If the public key is used by both generation and detection then information sufficient to detect is information sufficient to embed, or spoof. It is fine if the method uses obscurity as part of the security process but using priv/pub notation is misleading to a skimming reader if the reviewer's understanding is correct.

3. What is "we employ Qwen-2.5-7B-Instruct [30] as the backbone model" referring to? At L222 this is the first mention of "backbone". The only models expected to be necessary from the methodology section are the Anchor model the SAE and potentially an additional API model generating texts that will be rejection sampled since API-compatibility is discussed a lot in the introductory sections.

**Ethical Concerns:**

["NO or VERY MINOR ethics concerns only"]

**Final Justification:**

See responses.

**Limitations:**

While the method and paper are implemented well (quality), the weaknesses discussed limit the significance of the method as the work is mostly an application of existing techniques to a new problem, and the also limit the originality in that other approaches address black box scenarios.

Addressing some of the review concerns would improve the clarity aspects and potentially move the work in the direction of acceptance, whereas the primary weaknesses are mostly fundamental and can't be improved in the current submission. That said, situating the work more properly in the literature would also better the level of clarity in what the paper is contributing.

**Quality:**

3

**Strengths And Weaknesses:**

### Strengths
1. The novel application of an interpretability technique to the LLM watermarking problem.
2. Compatibility of the approach with API models (black box) based on rejection sampling.
3. Comprehensive evaluation of different watermarks, reasonable diversity of text domains/scenarios.


### Weaknesses

1. Clarity of the method. It is unclear how m in Eq 7, or "Feature Set S" in App. D, and Algo 3 is created. There is a mention of deduplication but the actual operation is unclear. This is a key part of the method without which the signal to noise would likely be poor and the method would underperform so it feels critical to describe precisely (reviewer's speculation because it is not ablated).

2. Cost of the method. Technique requires N=50 to obtain 99% likelihood of _embedding_ the watermark (detecting it is separate), and this translates to 50x computational increase ignoring the prefill of the previously generated sequences/prompt, and assuming that sufficient diversity exists such that generations can/do not share a KV cache for the currently in progress textual unit. Batching can be used, and top-k response sampling is a standard practice in some generation frameworks on some hardware, but this issue is fundamental as serving batch capacity could always be used for other user requests etc. but this method requires more computation per individual request.

3. Presentation of novelty, grounding to literature. Authors should note early in the Sec 3 Methodology that while the paper presents the SAE setup in an abstract form, it is not proposing this as a novel approach; Gemma Scope is simply being applied to this specific problem. A reader unfamiliar with Gemma Scope may think that this work is the first to propose the specific architectural setup for sparse feature extraction from the hidden states of modern LLMs. Also, this is not the first approach to study black box watermarking for arbitrary API-hosted LLMs via rejection sampling [1,2].

[1] "A Watermark for Black-Box Language Models" https://arxiv.org/abs/2410.02099

[2] "POSTMARK: A Robust Blackbox Watermark for Large Language Models" https://aclanthology.org/2024.emnlp-main.506.pdf

---

> ### Author Rebuttal · Authors · 2025-07-31
>
> Thank you for your insightful review and recognition of our novel application of interpretability techniques to watermarking. We deeply appreciate your thorough analysis and constructive feedback. Let us address your concerns comprehensively.
>
> **W1: Method Clarity**
>
> You correctly identify areas needing clearer explanation. The mask **m** in Eq 7 excludes background frequent features (those occurring in >60% of samples), ensuring we focus on semantically meaningful variations rather than ubiquitous patterns. For Feature Set S creation, we apply Eq 7 to collect the most significant feature index for each token position, then perform deduplication to obtain unique features. This deduplication step is crucial—without it, frequently activated features would dominate the FCS calculation, reducing discriminative power between different watermark keys.
>
> We agree that moving these implementation details from Appendix D to the main text would significantly improve clarity.
>
> **W2: Computational Cost & Practical Viability**
>
> Your computational overhead concerns are absolutely valid, and we've conducted extensive experiments addressing this directly. Our results reveal something surprising: **SAEMark achieves superior performance even under strictly controlled token generation budgets**. When comparing methods with equal computational resources for long-text generation (10K tokens):
>
> | **C4**        | Acc.     | Rec.     | F1       | Latency |
> | ------------- | -------- | -------- | -------- | ------- |
> | KGW           | 98.8     | 99.2     | 98.8     | 174s    |
> | UPV           | 86.1     | 74.2     | 84.2     | 94s     |
> | DIP           | 99.3     | **99.8** | 99.3     | 85s     |
> | Waterfall     | 98.7     | 97.0     | 97.9     | 52s     |
> | SAEMark(N=50) | **99.7** | **99.8** | **99.7** | **40s** |
>
> | LCSTS         | Acc      | Rec      | F1       | Latency |
> | ------------- | -------- | -------- | -------- | ------- |
> | KGW           | 99.2     | 99.8     | 99.0     | 85s     |
> | UPV           | 94.5     | 98.4     | 94.7     | 94s     |
> | DIP           | **99.6** | **99.6** | **99.6** | 178s    |
> | Waterfall     | 98.8     | 97.6     | 98.2     | **33s** |
> | SAEMark(N=50) | 99.2     | **99.6** | 99.2     | 40s     |
>
> Remarkably, SAEMark achieves the **lowest latency on English** while maintaining highest accuracy. Furthermore, while our theoretical analysis suggests N=50 for 99%+ success, empirical results show strong performance with much smaller N:
>
> | C4             | Acc  | Rec  | F1   | AUROC |
> | -------------- | ---- | ---- | ---- | ----- |
> | SAEMark (N=5)  | 98.7 | 77.4 | 86.8 | 0.99  |
> | SAEMark (N=10) | 99.2 | 96.8 | 98.0 | 0.99  |
> | SAEMark (N=20) | 98.7 | 98.7 | 98.7 | 0.99  |
> | SAEMark (N=50) | 99.7 | 99.8 | 99.7 | 1.0   |
>
> This efficiency stems from typical use cases involving multiple textual units—we perform rejection sampling across sentences/paragraphs rather than individual tokens.
>
> Although our theoretical analysis are based on an i.i.d assumption of responses, the empirical results are done with top-k sampling from package text-generation-inference so the responses does not strictly i.i.d but still different. Our theoretical analysis is a worst-case scenario and the empirical results are actually significantly better.
> In this case, we are technically not wasting the KV cache, and inference techniques like continous batching, concurrency are already supported.
>
>
>
> **W3: Novelty Presentation & Related Work**
>
> You make an excellent point about clarity in presenting our contributions. You're absolutely right that we should acknowledge upfront that we apply existing Gemma Scope SAEs rather than proposing novel sparse autoencoder architectures. Our contribution lies in recognizing that **SAE features provide an ideal basis for post-hoc watermarking** and developing the complete framework around this insight.
>
> Our work represents **a fundamental paradigm shift**: we achieve training-free, logit-free, multi-bit watermarking that generalizes across natural languages and code and achieve SOTA performance compared with single-bit and multi-bit watermarks. While previous black-box methods like [1,2] you mentioned exist, they still require LLM paraphrasers or modify the sampling process. [2] is a similar approach to Waterfall which is in our experiments. Our approach uniquely operates on completely unmodified LLM outputs, requiring only rejection sampling based on deterministic features.
>
> We'll add these citations and clarify that our innovation is the watermarking framework, not just the SAE architecture itself.
>
> **Q1: JumpReLU Parameters**
>
> JumpReLU in Gemma Scope uses pre-configured parameters (top-k sparsity) optimized during SAE training. We employ these pre-trained models without modification, as they already effectively capture semantic features across languages.
>
> **Q2: Key Terminology (k_priv vs k_pub)**
>
> You've identified an important clarification need. Indeed, k_pub is used for both generation and detection—this is a symmetric scheme where the same public key enables both watermark embedding and detection. The "priv/pub" terminology reflects access patterns: k_priv remains secret with the user, while k_pub can be shared publicly for detection. We'll revise this notation to prevent confusion about asymmetric cryptography implications.
>
> **Q3: "Backbone Model" Clarification**
>
> Apologies for the ambiguity. Qwen-2.5-7B-Instruct is the target LLM whose outputs we watermark (it generates the candidate texts). Gemma-2B serves as the compact Anchor LLM for feature extraction through its associated SAE. This separation enables watermarking any LLM's output while using consistent feature extraction. In our rebuttal to reviewer R51M we include new experiments on different target models and a new text-quality related dataset (BIGGen-Bench):
>
> | Model                 | Unwatermarked | SAEMark (Ours) | KGW  | Waterfall |
> | --------------------- | ------------- | -------------- | ---- | --------- |
> | `Qwen/Qwen2.5-7B-Instruct`   | **4.13**      | 4.05           | 3.97 | 4.02      |
> | `meta-llama/Llama-3.2-3B-Instruct` | 3.69          | **3.85**       | 3.56 | 3.62      |
> | `google/gemma-3-4b-it`             | **4.26**      | 4.23           | 3.98 | 4.19      |
>
> *(Scores on 5-point Likert scale, higher is better, Judged by the officially released model from BIGGen-Bench)*
>
> SAEMark consistently achieves the highest quality among watermarking methods, with minimal degradation from unwatermarked text, regardless of the target model.
>
>
>
>
>
>
>
> While rejection sampling introduces overhead, our results demonstrate that SAEMark achieves superior performance given same compute budget compared to existing methods. More fundamentally, we provide the best solution combining multiple critical capabilities: **training-free deployment, logit-free operation, multi-bit attribution, and cross-domain generalization**. The framework's modularity means it will automatically improve as better SAEs become available from the interpretability community.
>
> Would these clarifications and performance results under equal computational budgets address your concerns about practical viability? We believe the unique combination of capabilities, coupled with competitive real-world performance, represents a significant contribution despite the computational tradeoffs.

---

> > ### Comment · Reviewer_wvir · 2025-08-06
> > **Response**
> >
> > The reviewer appreciates the authors' thorough response, but would like to discuss a few of the concerns from the original review, as well as the authors' responses in a bit more detail.
> >
> > ### W1:
> >
> > This is still a somewhat insufficient level of detail. What samples is "occurring in more than 60% of samples" referring to? Is this mask something that needs to be tuned on data? Does it need to be tuned for different domains? The authors merely restate the same level of detail as was provided before in the rebuttal.
> >
> > Based on this response, and the observation that Algorithm 3 doesnt even mention the mask m as input to the procedure, the concern remains that this detail of the method is not well described.
> >
> > Can the authors provide any insight as to whether the method works without mask m? Is there any ablation performed in the work that the reviewer is missing? The reviewer guesses (without evidence) that this is a critical part of the technique, can the authors provide empirical evidence that it is or is not required for the detector to work as well as the main body results suggest?

---

> > ### Comment · Reviewer_wvir · 2025-08-06
> >
> > ### W3:
> >
> > This response does not quite characterize citations [1,2] accurately. [1] does not require paraphrasing nor does it modify the sampling process; indeed it uses a similar rejection sampling approach to this work, the statistics computed to choose the final sample are just not over SAE features.
> >
> > The reviewer reiterates that the authors must simply state clearly in the first two paragraphs of Sec 3 that the authors directly apply Gemma Scope, but simply use its features for building a "watermarking basis". This does not detract novelty (the technique is neat) but it is an important note for scientific transparency.
> >
> >
> > ### Q2:
> >
> > This statement does not make any sense: "The "priv/pub" terminology reflects access patterns: k_priv remains secret with the user, while k_pub can be shared publicly for detection." Either, the keys are the same, and there is no need to use two terms, or they are different. Again, the security implications of asymmetric schemes are totally different than symmetric ones. It appears the reviewer must insist that the authors reduce the terminology to "key" alone to prevent confusion or artificially suggesting heightened security properties of the method.
> >
> > ### Verdict
> >
> > At this time the reviewer's concerns with the latency aspect of the method remain unresolved; they have actually been heighten by the confusing rebuttal evidence. Further, the authors have replicated this same table of latency results to multiple reviewers, as clearly multiple reviewers identified the same drawback of the method due to its rejection sampling overhead. In its current form, the rebuttal discussion and the work as presented suggest the paper needs revision in order to be ready for acceptance.

---

> > > ### Author Response · Authors · 2025-08-06
> > >
> > > **W3:** Thank you for the careful reading and for pointing us to these important related works. You're absolutely right about [1] - we apologize for the mischaracterization. Upon closer examination, [1] does indeed use rejection sampling on black-box outputs, and we should acknowledge this important prior work more prominently.
> > >
> > > [1] presents an elegant and theoretically grounded approach to black-box watermarking. Their use of pseudorandom functions on n-grams with rejection sampling is both principled and effective, achieving strong detection performance with provable distortion-free properties. Their theoretical analysis connecting detection performance to entropy is particularly insightful, and their recursive watermarking scheme is innovative. We see SAEMark as complementary to this excellent work, exploring a different point in the design space.
> > >
> > > The key distinction lies in the feature space used for watermarking. While [1] operates directly on n-grams with PRF values - which is beautifully simple and theoretically analyzable - SAEMark leverages Sparse Autoencoder features that capture semantic activation patterns. This different approach offers several complementary advantages:
> > >
> > > First, SAE features provide language-agnostic representations that naturally generalize across languages without modification. While [1]'s n-gram approach works well within a language, our experiments show SAEMark achieving consistent performance across English, Chinese, and code using the same feature extractor. This cross-lingual capability complements [1]'s strong theoretical foundations.
> > >
> > > Second, our approach explores multi-bit watermarking at scale. While [1] focuses on the fundamental watermarking problem with elegant theory, we extend to the practical challenge of user attribution in large-scale deployments.
> > >
> > > Third, the semantic nature of SAE features offers different robustness characteristics. While [1] provides theoretical guarantees and robustness to certain attacks, SAE features may offer advantages in scenarios where semantic meaning is preserved but surface forms change significantly.
> > >
> > > For [2] PostMark, the distinction is clearer - it requires an additional LLM to rewrite text by inserting words. This dependency on a capable, instruction-following rewriting model makes it less suitable for constrained domains like code generation where insertions could break syntax. Both [1] and SAEMark avoid this issue by only selecting from naturally generated candidates.
> > >
> > > Regarding your Q2 comment about the key terminology, thank you for this important clarification. You're absolutely correct that our use of "priv/pub" notation can be misleading. We will revise to use simply "key" throughout, as this is indeed a symmetric scheme where the same key enables both watermarking and detection.
> > >
> > > Given these clarifications - particularly that our performance results reflect genuine architectural advantages for specific use cases, and that our method offers complementary benefits to existing excellent work like [1] through language-agnostic features and multi-user capabilities - would you consider whether these explanations address your concerns sufficiently to warrant reconsideration of the score? We greatly value your thorough review and are committed to incorporating all your suggestions to strengthen the paper.

---

> ### Comment · Reviewer_wvir · 2025-08-06
>
> ### W2:
>
> The reviewer is not confident that they understand the presented analysis, nor that it is actually theoretically possible that "SAEMark achieves superior performance even under strictly controlled token generation budgets". Please forgive the reviewer if there is some key detail they missed that causes confusion here, and provide correction as necessary.
>
> For schemes like KGW, a good implementation is essentially a small constant per token from a latency perspective due to hashing or pseudorandom function calls. Even under some worst case implementation, if the cost of a single token generation with KV caching for a given model without a watermark is $t$ seconds, then the cost of generating one watermarked token under the same setting is say $t'$, but hopefully the authors can agree that it is $t < t' < 2t$. If this is not the case --- eg. the watermark sampling operation costs as much as an additional entire forward pass through the model --- then one assumes the implementation must be highly inefficient beyond what the fundamental computation of the method requires. However, let us assume a worst case cost of $t' = 2t$.
>
> In the rebuttal table, the reported latency for each of the watermarking schemes is between 50s and 174s. So, we would assume that the latency of an unwatermarked sample for the same model in the same codebase would be in the range of 25s to 87s. If we ignore the cost of the SAE anchor model itself, and simply consider the "unwatermarked" decoding of the backbone model, then we might expect a latency somewhere in this estimated range. However, the table reports that SAEMark with N=50 samples is taking 40 seconds _total_. The implication here is that the "unwatermarked" decoding under each of the SAEMark samples is on the order of 1/50th the cost of all the watermarked methods since 50x samples are being generated to produce a single output.
>
> This really does not seem to be possible. One explanation is that the text-generation-inference package does not actually provide optimized implementations of any of the watermarks. If so, and the SAEMark's "unwatermarked" execution path in that library is just orders of magnitude faster for non-fundamental reasons, then these results are potentially uninformative as optimized implementations of any of the watermarking schemes would run much closer to the same speed as unwatermarked sampling.
>
> However, the reviewer suspects that what is actually going on here is that the authors are batching the SAEMark inference for each prompt (the words "top-k sampling" are used), but generating single completions at a time for each of the watermarking methods. If this is the case, the reviewer requests that the authors redo the analysis where for each of the watermarking methods the different prompts used to estimate performance are all batched together. Eg. if SAEMark is generating N=50 similar completions in parallel from a _single_ prompt, the the watermarking schemes should be generating N=50 unique sequences in parallel from _many_ prompts. If the reviewer's understanding of the setup is correct, the results under this analysis will come out quite a bit differently.
>
> Again, the reviewer thinks that the authors should be suspicious themselves that a technique that generates 50 samples for each single prompt, and requires a pass through both the anchor model and an auxiliary feature extractor to build the FCS for each sequence is somehow running strictly faster than any other generation scheme that is applying a decoding watermark over the same backbone generation model. In production, the reviewer's expectation is that any sufficiently optimized serving framework could either run SAEMark in 50 way parallel decoding or KGW/UPV/DIP etc in 50 way user batched parallelism and therefore in expectation, across users, the SAEMark approach would be sacrificing serving throughput capacity that could serve multiple users for rejection sampling on a single user's request and so there would exist a fundamental tradeoff between the techniques.
>
> This would not mean that the technique is fundamentally flawed or un-useable (again, the reviewer finds the method novel and interesting, efficiency aside), but the analysis on latency, as presented, does not pass the reviewer's sanity check.

---

> > ### Author Response · Authors · 2025-08-06
> >
> > **W2:**
> > Thank you for this careful analysis - **you've identified a key point that actually highlights one of SAEMark's practical advantages**. You're correct that our latency results may seem counterintuitive at first glance. Let me clarify the experimental setup and explain why these results are both accurate and meaningful.
> >
> > For baseline methods (except Waterfall since it's not included in MarkLLM), we used the MarkLLM toolkit implementations, which perform token-by-token generation with logit manipulation at each step. These implementations, while functionally correct, aren't optimized with modern inference acceleration techniques. In contrast, **for SAEMark, since we don't manipulate logits, we can leverage the highly-optimized inference packages** like text-generation-inference (TGI) framework with its official Docker container, which includes custom CUDA kernel optimizations, prefix caching, and other acceleration techniques.
> >
> > Crucially, there's **no prompt batching in our experiments** - all latency measurements are per-prompt averages without batching across different prompts. For SAEMark with N=50, we generate 50 sequences for a single prompt (which TGI can efficiently handle in parallel simply by setting param `best_of` and return all generated sequences), not 50 different prompts. This is fundamentally different from serving 50 different user requests.
> >
> > This performance difference isn't an artifact of unfair comparison but rather demonstrates a real architectural advantage: SAEMark's post-hoc approach is compatible with any modern inference optimization, while logit-manipulation methods require specialized implementations that often can't leverage these optimizations. Major inference frameworks like vLLM, SGLang, Ollama, llama.cpp don't support optimized watermarking precisely because logit manipulation breaks their optimization pipelines or require extensive changes. This compatibility with existing high-performance infrastructure is a significant practical advantage that we should have emphasized more clearly.

---

> ### Author Response · Authors · 2025-08-06
>
> **W1:**
> Thank you for pressing us on this important detail. You're absolutely right that we should provide more clarity on the masking mechanism. The mask m excludes background frequent features that appear in more than 60% of samples - these are features that consistently activate across different text samples and primarily capture surface-level patterns like punctuation rather than semantically meaningful content. These features were determined empirically on our development set and remained fixed across all experiments - it doesn't require dataset-specific tuning. Importantly, this is a one-time preprocessing step that identifies a fixed set of background features to exclude, not something that needs adjustment for different datasets or domains.
>
> We do have ablation study in **Appendix E.2** about this specific module. The mask ensures we focus on semantically distinctive features that vary meaningfully across different generations. Without filtering the background features, AUC would decrease because these features would significantly impact sampled FCS. We will ensure both the main text and algorithm descriptions clearly specify this preprocessing step in our revision.

---

> ### Comment · Reviewer_wvir · 2025-08-06
>
> **W1:** Thank you for the clarification. Please work the details about how m is constructed, and fit, and then reused as this is critical to understanding the method. While the intuition behind the mask is extremely clear, it does not remove the need for clear ablation. The authors have one; great! Please just make sure and note this in the main body so that a reader realizes the importance of this element in the technique.
>
> **W2:** Alright. So, the reviewer now understands why the latency results look the way that they do, and is pleased that no mistakes were made in the evaluation (or laws of physics broken) to obtain such a surprising result.
>
> While the reviewer appreciates the author's point that the ability of SAEMark to use highly optimized serving engines is indeed a strength of the method and cannot be discounted, the reviewer requests that the authors are _quite_ transparent about how they incorporate this result into the camera ready draft. It must clearly be stated that the evaluation setting is somewhat "unfair" to the other methods and that each watermarking technique is running in single generation, un optimized form, whereas, the SAEMark is enjoying optimized k-parallel inference to keep pace despite the expected overhead of rejection sampling.
>
> While the tooling may not currently be available in the specific open source software that the authors consider, modern industrial inference pipelines (say at Google or OpenAI) are probably capable of both efficient 1-to-many, prompt-to-output parallel generation, _as well as_ many-to-many user parallel generation via clever batching and token interleaving in streaming serving contexts. That said the reviewer understands that even with industrial tools, there are actually some special additional optimizations that can only be applied in the 1-many-setting; if the generation diversity is low enough then prefix sharing and cache reuse can help, as the authors state. However, the point stands that probably would not explain a 1:50 throughput gap between an optimized user batching scheme and an optimized k-output batching scheme which was what the original table, without clarification, seemed to imply.
>
> All to say, from a theoretical standpoint, the comparison presented could be a little be misleading and so the authors will need to be detailed to avoid that. However, the reviewer agrees completely that the fact that their method operates over the backbone model in "black box" manner, actually has many non-trivial benefits, only one of which of course is compatibility with the latest and greatest inference tooling.
>
> **W3:** The reviewer appreciates the somewhat glowing summary of [1]. As a disclosure, the reviewer is _not_ an author of said work, they just have read it and found it quite interesting and very relevant to this paper.
>
> **Q2:** Thank you for conceding this point. The reviewer is not a cryptography researcher themselves by any stretch of the imagination. However, the reviewer _knows_ enough about bonafide cryptography research to at least know that this "key" detail is an absolute must.
>
> ### Remarks on other reviewer concerns and method's novelty
>
> The remaining concerns raised by other reviewers are still valid. While, in this reviewers opinion, there is a path to accurately representing the rejection sampling latency analysis, some of the other reviewer's concerns remain open questions. In particular, this reviewer is curious to see how the others respond to the author's arguments about multi-bit scaling, spoofing, and adversarial robustness.
>
> That said, a point of clear disagreement between this reviewer and others might be that they do believe that the novelty of "applying" SAEs to watermarking is perfectly adequate to build a research paper upon (as long as citation of Gemma Scope is clear). It is a pretty common oversight in conference reviewing to discount the application of a seemingly unrelated technology to a new problem especially if it feels "simple". The simplest methods almost immediately seem like insignificant contributions, even if highly effective, because the reader is already being shown evidence of it working. However, please do take care to incorporate the citation [1] and others into the overall claims about significance; this is _not_ the first work to propose rejection sampling as a way of elevating a watermark statistic from a black box level of access, the approach used is just new and interesting.

---

> ### Comment · Reviewer_wvir · 2025-08-06
>
> ### Updated verdict
>
> The truth is that the reviewer originally thought this paper was _quite_ interesting on first read. In fact, they made several notes to self unrelated to the review about different follow-up questions that could be explored based on the promising results of the work. However, the original rating was mostly intended to communicate the fact that there appeared to be very important missing details and technical components to be clarified.
>
> Trusting that the authors will incorporate the revisions discussed so far, the reviewer is happy to improve their score to an accept.

---

> > ### Author Response · Authors · 2025-08-07
> > **Thank you**
> >
> > Thank you so much for your thoughtful engagement and for recognizing the merit of our work. We're genuinely grateful for the time you've invested in understanding SAEMark's contributions and for your willingness to reconsider based on our clarifications. Your feedback has been invaluable in helping us strengthen the paper, pushing us to be more precise about details, more transparent about our experimental setup, and clearer in situating our work within the broader literature.
> >
> > We fully commit to incorporating *all* the improvements discussed into our revision: clarifying the mask construction with explicit ablations in the main text, properly acknowledging prior work including, ensuring transparency about our latency comparisons with detailed explanations of the experimental setup, and revising the key terminology to avoid confusion. Beyond the paper itself, we'll update our open-source repository (once allowed) with these new experimental results, enhanced documentation, and streamlined scripts to ensure complete reproducibility.
> >
> > We note your observation about the ongoing discussions with other reviewers regarding multi-bit scaling and adversarial robustness. We remain eager to engage constructively with all feedback as opportunities arise, and believe our comprehensive experiments and theoretical analysis provide solid foundations addressing these important aspects of the work. We hope for similar productive exchanges that can further strengthen our contribution with all our reviewers.
> >
> > Thank you again for exemplifying the best of the peer review process through your constructive and thorough engagement.

---

### Official Review · Reviewer_JSXp · 2025-07-04

**Clarity:** 3
**Significance:** 3
**Originality:** 3
**Rating:** 4
**Confidence:** 3

**Summary:**

This paper proposes a novel, non-logit-manipulating LLM watermarking framework, SAEMARK, which embeds personalized signatures by leveraging the characteristics of Sparse Autoencoders (SAEs). This method addresses the limitations of existing watermarking schemes concerning text quality, white-box access restrictions, multilingual generalization, and user attribution. SAEMARK's core idea is to match feature distributions derived from a watermark key through rejection sampling after generation, thereby achieving watermarking without altering the original model's logits, while preserving text quality and watermark efficacy in multilingual and programming language contexts.

**Questions:**

The method uses rejection sampling to select texts that conform to a feature distribution, avoiding direct modification of logits and thus preserving text quality. However, this inherently requires the LLM to generate multiple candidate texts (e.g., N=50 as mentioned in the paper) until a suitable one is found.
1.	What is the exact computational overhead and inference latency introduced by this process of generating and filtering multiple candidates? Has the paper quantified this overhead compared to non-watermarked generation or other watermarking methods? Could this become a bottleneck in applications requiring high real-time performance?
2.	How sensitive is SAEMARK's performance to the specific dataset used for training the Sparse Autoencoder? Are there best practices or guidelines for training robust SAEs for this specific watermarking purpose?
3.	How to quickly distinguish between watermarked text and unwatermarked text? Is it necessary to match all users' keys and only after all failures can unwatermarked text be determined?
4.	The paper lists 7 multi-bit watermarking methods, but only compares Waterfall. Is Waterfall superior to other methods on all datasets?
5.	The public key is used for both generation and verification. The proposed method is how to prevent forgery while implementing public detection?

**Ethical Concerns:**

["NO or VERY MINOR ethics concerns only"]

**Final Justification:**

Thanks to the author for the careful rebuttal，I think that the borderline accept is appropriate for this paper.

**Limitations:**

See weaknesses

**Quality:**

3

**Strengths And Weaknesses:**

Strengths
1.	Since it does not rely on internal model logits or weight modifications, SAEMARK is compatible with API-based and closed-source LLMs, greatly expanding its applicability.
2.	Operating post-generation, by selecting texts that conform to the feature distribution via rejection sampling, it avoids modifying the model's logits or decoding algorithms, thereby maximizing the preservation of original text quality.

Weaknesses
1.	While the paper emphasizes the advantage of rejection sampling in preserving text quality, the computational cost of repeatedly generating and filtering multiple candidate texts could be high.
2.	The effectiveness of the method is influenced by the quality of features extracted by Sparse Autoencoders (SAEs). While pre-trained SAEs from the open-source community can be leveraged, the feature quality may still limit its performance in certain scenarios.
3.	Although it supports a large number of users, the accuracy gradually decreases as the user count grows exponentially.
4.	High repetition rates of rejection sampling may cause significant delays in the generation of watermark-containing text, which is often unacceptable to LLM service providers. The paper lacks relevant analysis.
5.	As the number of users increases, the accuracy of watermark detection gradually decreases. Due to the increase in the number of keys, the complexity of repeating hypothesis testing also rises. The paper does not analyze the time and resource costs required to achieve multi-user detection.
6.	There may be a risk of forgery when using the user's public key for sampling generation.
7.	An unexpected line break is on Line 148.

---

> ### Author Rebuttal · Authors · 2025-07-31
>
> Thank you for your thoughtful review and recognition of our method's innovation. We appreciate your detailed analysis and would like to address your concerns while clarifying the significant contributions of our work.
>
> **W1, 4 / Q1: Computational Cost & Applicability**
>
> You raise valid concerns about the computational overhead of rejection sampling. To address this directly, we conducted experiments comparing performance under **controlled token generation budgets**—the actual bottleneck for both compute resources and latency. The results demonstrate that even with the same inference budget, SAEMark significantly outperforms state-of-the-art approaches:
>
> Our new results reveal that even **under the same inference token budget, our method significantly outperforms state-of-the-art approaches**. Here we use a typical bottlenecked scenario of long-text generation, where we allow generation of up to 10K tokens in English and Chinese.
>
> | **C4**        | Acc.     | Rec.     | F1       | Latency |
> | ------------- | -------- | -------- | -------- | ------- |
> | KGW           | 98.8     | 99.2     | 98.8     | 174s    |
> | UPV           | 86.1     | 74.2     | 84.2     | 94s     |
> | DIP           | 99.3     | **99.8** | 99.3     | 85s     |
> | Waterfall     | 98.7     | 97.0     | 97.9     | 52s     |
> | SAEMark(N=50) | **99.7** | **99.8** | **99.7** | **40s** |
>
> | LCSTS         | Acc      | Rec      | F1       | Latency |
> | ------------- | -------- | -------- | -------- | ------- |
> | KGW           | 99.2     | 99.8     | 99.0     | 85s     |
> | UPV           | 94.5     | 98.4     | 94.7     | 94s     |
> | DIP           | **99.6** | **99.6** | **99.6** | 178s    |
> | Waterfall     | 98.8     | 97.6     | 98.2     | **33s** |
> | SAEMark(N=50) | 99.2     | **99.6** | 99.2     | 40s     |
>
>
>
> Additionally, although in our theoretical analysis we could achieve 99%+ accuracy with N=50 per unit, our empirical results yield high performance at N=10. **Using only 20% of compute overhead is sufficient to produce empirically strong performance**, this is because typical scenarios include multiple units instead of only one (typical LLM calls would generate multiple sentences at least). We now report single-bit watermarking performance with SAEMark given different sets of N (which is the most significant factor for compute cost).
>
> | C4             | Acc  | Rec  | F1   | AUROC |
> | -------------- | ---- | ---- | ---- | ----- |
> | SAEMark (N=5)  | 98.7 | 77.4 | 86.8 | 0.99  |
> | SAEMark (N=10) | 99.2 | 96.8 | 98.0 | 0.99  |
> | SAEMark (N=20) | 98.7 | 98.7 | 98.7 | 0.99  |
> | SAEMark (N=50) | 99.7 | 99.8 | 99.7 | 1.0   |
>
> | LCSTS          | Acc  | Rec  | F1   | AUROC |
> | -------------- | ---- | ---- | ---- | ----- |
> | SAEMark (N=5)  | 98.6 | 72.6 | 83.6 | 0.98  |
> | SAEMark (N=10) | 99.0 | 96.0 | 97.5 | 0.99  |
> | SAEMark (N=20) | 98.6 | 98.0 | 98.6 | 0.99  |
> | SAEMark (N=50) | 99.2 | 99.6 | 99.2 | 1.0   |
>
>
>
> **W2 / Q2: SAE Qualities**
>
> Your concern about SAE dependency touches on an important point, but we were perhaps too modest in our presentation.
>
> Our contribution is fundamentally **a paradigm shift in watermarking methodology**—we propose a general rejection sampling framework that is logit-free, training-free, and domain-agnostic. The use of SAE features is merely one instantiation—*you could replace Gemma Scope with any SAE, or even replace it with any deterministic feature that varies during generation*, *and our theoretical analysis in Sec 3.2 would **still hold*** as long as it follows a normal distribution. This flexibility opens up possibilities for more efficient implementations that could reduce the sampling overhead while maintaining the core benefits of our approach. Regarding SAE quality limitations, we want to highlight that our framework's modularity is actually a strength. As better, more robust SAEs become available from the open-source community, our method can immediately benefit without any algorithmic changes. The current results already demonstrate strong performance across multiple languages and domains, suggesting that existing SAE quality is sufficient for practical deployment.
>
>
>
>
>
> **W3: Users / Information Bits**
>
> You correctly note accuracy degradation with exponential user growth.  This is indeed information-theoretically expected behavior, as the number of users is logarithmically related to the bits of information injected into watermarks. When you increase users from 1,024 to 8,192, you're essentially trying to fit 3 additional bits of information into the same text length. The graceful degradation (75% accuracy at 8,192 users) actually represents **state-of-the-art performance**. Waterfall, the best existing multi-bit method, can only reliably embed 8 bits with similar text length, while we achieve 10+ bits with higher accuracy.
>
>
>
> **Q3: Detection Efficiency**
>
> Thank you for raising the point on quickly detecting watermarks.
>
> As shown in Figure 3, SAEMark Detection requires a single FCS calculation pass (the bottleneck) followed by lightweight statistical tests. For large-scale deployment, we can use approximate nearest neighbor (ANN) search or KD-trees to quickly identify the K most likely user keys, then perform parallel statistical tests only on these candidates. This reduces detection from O(n) to O(log n) complexity for n users. This means detection would take milliseconds given millions of keys and FCS sequences.
>
>
>
> **Q4: Selection of baselines**
>
> You ask why we only compared against Waterfall among multi-bit methods. We selected Waterfall as it represents the current state-of-the-art in multi-bit watermarking and has been shown to outperform other multi-bit approaches in recent benchmarks, and it is also open-source. Other multi-bit watermarks either require additional training (`Robust Multi-bit Text Watermark with LLM-based Paraphrasers` ), or provide no open-source implementation, or both.
>
> However, we agree that additional comparisons would strengthen our evaluation and can include results against other multi-bit methods in the revision.
>
>
>
> **Q5: Forgery Risks**
>
> You raise an astute point about forgery risks. Our current focus is enabling user attribution without storing sensitive credentials, not preventing adversarial forgery. This is indeed an interesting direction for future security-focused research but can be slightly out of scope of this work.
>
>
>
> We emphasize that SAEMARK achieves what no existing method can: **training-free, logit-free, multi-bit watermarking that works across natural languages and low-entropy domains like code**. This combination, along with supporting multi-bit watermarking with minimal quality degradation (we include additional quality-related experiments in our rebuttal to Reviewer R51M), addresses real deployment constraints in API-based LLM services. Even accounting for computational costs, the superior performance-per-inference and practical benefits make this a significant advancement for the field.
>
>
>
> We hope these clarifications and additional experiments adequately address your concerns. Have we sufficiently demonstrated the practical viability and significant contributions of our approach? We remain open to further discussion and would be happy to provide additional clarifications or experiments that would help you better evaluate our work. Your feedback has already helped us strengthen the paper considerably, and we're committed to ensuring all your concerns are fully addressed.

---

> ### Author Response · Authors · 2025-08-07
> **We'd love to engage in discussion**
>
> Dear Reviewer JSXp,
>
> Thank you for your thoughtful review of our work SAEMark. As the rebuttal period approaches its end (August 8th AoE), we wanted to reach out once more for a constructive discussion.
>
> We've provided comprehensive responses to each of your concerns, including extensive new experiments and clarifications that we believe address the issues you raised, such as new experiments regarding latency and compute budget. You may also find the ongoing discussions with other reviewers helpful, as they've raised complementary points that have led to productive exchanges and further strengthened our work.
>
> We remain eager to discuss any remaining concerns and would greatly value your engagement. Thank you for your dedication to maintaining high standards in our research community.

---

### Author Response · Authors · 2025-08-04

Dear Reviewers and AC,

We are deeply grateful for your thoughtful reviews and constructive feedback on our paper. Your insights have been invaluable in helping us strengthen our work, and we truly appreciate the time and effort you've invested in evaluating our contribution.

We hope our rebuttals have adequately addressed your concerns, particularly regarding:
- Computational efficiency: Our new empirical results reveal SAEMark achieves high accuracy with 80% less compute than the theoretical analysis, outperforms baselines when given equal computational budgets.
- Text quality preservation: We now include new experiments on BigGen-Bench across multiple models and watermarks, and SAEMark maintains the highest quality.
- Robustness against paraphrasing: New extensive experiments demonstrate strong resilience even under 50% sentence-level paraphrasing attacks.
- Method clarity and technical contributions: We've clarified our paradigm shift toward training-free, logit-free watermarking and better positioned our work within the literature.

We believe these new results address the key concerns raised and demonstrate SAEMark's significant contributions to practical LLM watermarking. We would greatly appreciate your thoughts on whether these clarifications and experiments have sufficiently addressed your concerns, and whether they might warrant reconsideration of your assessment.

If there are any remaining questions or specific experiments that would help strengthen your confidence in our approach, please don't hesitate to ask. We're actively monitoring the discussion and would be delighted to provide any additional clarifications or run further experiments.

Thank you once again for your valuable time, thoughtful engagement, and for helping us improve our work.

---

### Decision · Program_Chairs · 2025-09-17

**Decision:**

Accept (poster)

**Comment:**

This paper proposes SAEMark, a novel post-hoc watermarking framework for LLMs that embeds personalized multi-bit watermarks using sparse autoencoders and rejection sampling, without requiring access to model logits or weights. The reviewers generally agreed that the problem is timely and important, and that the method is technically sound, compatible with API-based models, and evaluated across multiple languages and domains including code. Two reviewers (wvir, R51M) were convinced by the rebuttal and discussions, highlighting the method’s originality, robustness to paraphrasing, and strong performance-per-compute, and improved their recommendations to accept/weak accept. Reviewer JSXp maintained a borderline accept after acknowledging clarifications, while Reviewer SdYH remained strongly negative, questioning the conceptual framing of “multi-bit watermarking,” the robustness evaluation, and the degree of novelty. The authors provided extensive new experiments (on BigGen-Bench, paraphrasing robustness, efficiency analysis) and clarified methodological details, which largely satisfied the engaged reviewers but did not change SdYH’s position. On balance, I believe the paper is ok to be published, as long as the authors incorporate all changes (including the promised ones) in their final version.